# GRAPH NEURAL NETWORKS FOR EDGE SIGNALS: ORIENTATION EQUIVARIANCE AND INVARIANCE

**Dominik Fuchsgruber[1], Tim Poštuvan[2], Stephan Günnemann[1], Simon Geisler[1]**
[1]Department of Computer Science & Munich Data Science Institute, TU Munich [2]EPFL
{d.fuchsgruber,s.guennemann,s.geisler}@tum.de, tim.postuvan@epfl.ch

## ABSTRACT

Many applications in traffic, civil engineering, or electrical engineering revolve around edge-level signals. Such signals can be categorized as inherently directed, for example, the water flow in a pipe network, and undirected, like the diameter of a pipe. Topological methods model edge signals with inherent direction by representing them relative to a so-called *orientation* assigned to each edge. They can neither model undirected edge signals nor distinguish if an edge itself is directed or undirected. We address these shortcomings by (i) revising the notion of *orientation equivariance* to enable edge direction-aware topological models, (ii) proposing *orientation invariance* as an additional requirement to describe signals without inherent direction, and (iii) developing EIGN, an architecture composed of novel direction-aware edge-level graph shift operators, that provably fulfills the aforementioned desiderata. It is the first work that discusses modeling directed and undirected signals while distinguishing between directed and undirected edges. A comprehensive evaluation shows that EIGN outperforms prior work in edge-level tasks, improving in RMSE on flow simulation tasks by up to 23.5%.

## 1 INTRODUCTION

Most research on Graph Neural Networks (GNN) research has focused on node- or graph-level tasks (Wu et al., 2021; Kipf & Welling, 2017; Hamilton et al., 2017; Velickovic et al., 2018; Gasteiger et al., 2020; AlQuraishi, 2021) while edge-level problems remain underexplored. Edge-level signals describe the properties of *existing* edges and are either the features, hidden representations of a GNN, or the targets. They naturally arise in applications involving traffic and many areas of engineering (e.g. electric circuits (Dörfler et al., 2018) or hydraulics (Garzón et al., 2022; Herrera et al., 2016)).

Signals on edges come in two different modalities: (a) They can have an inherent orientation, like the water flow in a pipe network or traffic flow on streets. We call them *orientation-equivariant* signals as they are naturally expressed as scalar values relative to a chosen orientation as reference (see Section 2). (b) The other signal modality has no intrinsic direction, like pipe diameter or speed limits. We refer to such signals as *orientation-invariant*. Similar concepts also exist in continuous domains, e.g. a scalar field assigns a single value to each point in space, while a vector field describes magnitude and direction.

At the same time, edges themselves may be directed or undirected. Examples that induce directed edges include valves in water networks, one-way roads in street networks, or diodes in electrical circuits. Often, in these applications, directed edges prohibit (to a large extent) a signal that is oriented against the edge direction. Such constraints can have a big impact on the targets. For example, imagine a one-way street that forces cars to take a detour.

There are two relevant streams of work that, however, fall short of either representing orientation-equivariant or orientation-invariant signals. (a) One strategy is to map between orientation-invariant edge-level signals using Line Graphs. (b) The other line of work relies on Algebraic Topology (Schaub et al., 2021; Ebli et al., 2020; Bunch et al., 2020) to cope with orientation-equivariant signals. Topological models define an arbitrary reference direction for each edge, a so-called *orientation*, and use positive values to indicate that an orientation-equivariant signal aligns with this reference orientation whereas negative values indicate misalignment (see how in Figures 1a

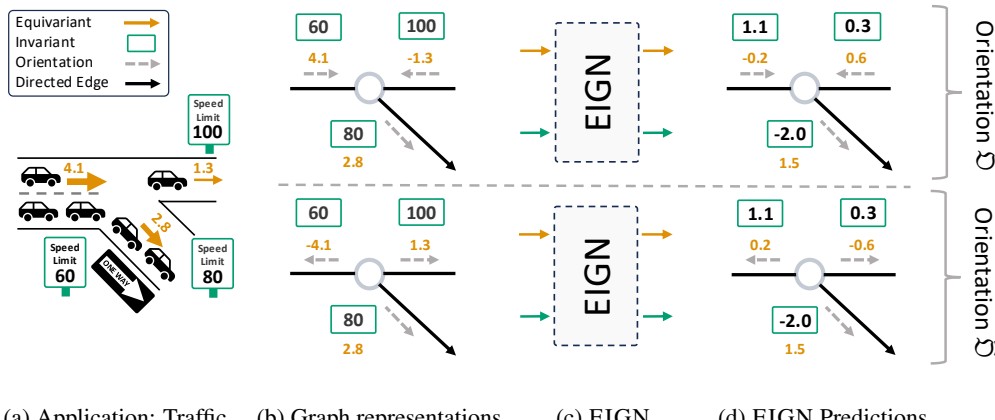

(a) Application: Traffic    (b) Graph representations    (c) EIGN    (d) EIGN Predictions

Figure 1: EIGN models an arbitrary combination of orientation-equivariant and -invariant edge-level inputs or targets. In this example, the car flow is equivariant and represented relative to two different (top and bottom) arbitrary direction-consistent orientations $\mathfrak{O}$ and $\hat{\mathfrak{O}}$ (notice the sign of equivariant signals), while speed limits are invariant. EIGN makes consistent predictions for $\mathfrak{O}$ and $\hat{\mathfrak{O}}$: It outputs the same invariant signals while the sign of equivariant outputs is determined by the orientation.

and 1b, the signal $1.3$ is represented as $-1.3$). For undirected edges, there is no preferred reference orientation. Consequently, a common requirement for topological models is to be *orientation-equivariant* (Roddenberry et al., 2021): The sign of the (orientation-equivariant) signal must change together with the orientation of the corresponding edge (see Figure 1). Orientation equivariance is a property of a topological model that allows it to deal with orientation-equivariant signals.

Many applications have both orientation-equivariant and invariant features with an arbitrary combination of orientation-equivariant and invariant targets. Moreover, real scenarios are often only accurately modeled if using both undirected and directed edges. However, Line Graph approaches ignore the properties of orientation-equivariant signals entirely, while topological approaches treat every signal as orientation-equivariant and can not distinguish edge direction. Arguably, the inductive biases of prior methods render them ineffective in a large range of applications, which is consistent with our experimental findings.

We address these shortcomings and are the first to model edge-level tasks with orientation-equivariant and -invariant signals on graphs with directed and undirected edges. Our key contributions are:

a) **Desiderata.** We formalize suitable desiderata: (i) *Joint orientation equivariance* for orientation-equivariant signals, and (ii) *joint orientation invariance* for orientation-invariant signals.

b) **EIGN: A general-purpose edge-level topological GNN.** EIGN provably fulfills these desiderata, leveraging novel direction-aware convolution operators. and a fusion operation between both modalities to model their interactions ( see Section 4.2).

c) **Benchmarking.** We devise a suite of challenging and diverse benchmarks covering synthetic and real-world tasks, including a novel dataset for electric circuits that requires dealing with orientation-equivariant and orientation-invariant signals on graphs with directed and undirected edges. EIGN outperforms prior work by reducing the RMSE up to 23.5% in our experiments.[1]

## 2 BACKGROUND

Algebraic Topology is a principled framework for representing orientation-equivariant edge signals (Roddenberry & Segarra, 2019; Roddenberry et al., 2021; Ebli et al., 2020). In the interest of notational clarity, we simplify many concepts by directly applying them to the edge domain and omit general topological definitions. For a comprehensive introduction, we refer to Schaub et al. (2021).

**Notation.** We consider edge-level problems on a graph $\mathcal{G} = (\mathcal{V}, \mathcal{E})$ with $n = |\mathcal{V}|$ nodes and $m = |\mathcal{E}|$ edges. We distinguish between: (i) directed edges, represented by ordered tuples $\mathcal{E}_\mathrm{D} \subseteq \mathcal{V} \times \mathcal{V}$, and undirected edges $\mathcal{E}_\mathrm{U} \subseteq \{\{u, v\} : u, v \in \mathcal{V}\}$, represented by unordered sets. (ii) edge-signals

---

[1]We provide our code at cs.cit.tum.de/daml/eign/

(inputs, hidden representations, outputs) that come with inherent direction as *orientation-equivariant* $\boldsymbol{X}_{\mathrm{equ}} \in \mathbb{C}^{m \times d_{\mathrm{equ}}}$ and *orientation-invariant* $\boldsymbol{X}_{\mathrm{inv}} \in \mathbb{C}^{m \times d_{\mathrm{inv}}}$ otherwise.

**Orientation.** While orientation-invariant edge signals can be represented by scalar values as they are, orientation-equivariant edge signals need to be defined relative to the orientation of the associated edge. To that end, topological methods define an arbitrarily chosen orientation $\mathfrak{O} : \mathcal{E} \to \mathcal{V} \times \mathcal{V}$ for every edge, also represented by ordered tuples. If an orientation-equivariant signal $x$ aligns with the orientation of its edge, we represent it with $x$ and $-x$ otherwise. Changing from orientation $\mathfrak{O}$ to another orientation $\hat{\mathfrak{O}}$ induces sign flips for all orientation-equivariant signals of edges whose orientation was flipped. We can represent this orientation change for signals with a diagonal matrix $\Delta_{\mathfrak{O},\hat{\mathfrak{O}}} \in \mathbb{R}^{m \times m}$ with entries $[\Delta_{\mathfrak{O},\hat{\mathfrak{O}}}]_{e,e} = 1$ if $\mathfrak{O}(e) = \hat{\mathfrak{O}}(e)$ and $-1$ otherwise.

Edge direction and orientation are different concepts. While the former is part of the given problem topology and may influence model predictions, the latter can be thought of as a basis in which orientation-equivariant edge signals are represented. For undirected edges, which orientation is chosen must not impact the problem itself. However, unlike previous work, we fix the orientation of *directed edges* to match their direction ($\mathfrak{O}(e_{\mathrm{D}}) = e_{\mathrm{D}}$) and, consequently, represent their orientation-equivariant signals relative to their direction. We refer to such orientations as *direction-consistent*. They can encode information about the direction of *directed* edges.

**Boundary and Laplace Operators.** For a given orientation $\mathfrak{O}$, one can define a boundary operator that maps signals on $m$ edges defined with respect to orientation $\mathfrak{O}$ to the domain of $n$ nodes. Reusing the analogy of water flow, the boundary operator sums all flow coming into a node and subtracts all outgoing flow. It can be represented by a matrix $\boldsymbol{B}_{\mathrm{equ}} \in \mathbb{R}^{n \times m}$:

$$[\boldsymbol{B}_{\mathrm{equ}}]_{v,e} = \begin{cases} -1 & \text{if } \mathfrak{O}(e) = (v, \cdot) \\ 1 & \text{if } \mathfrak{O}(e) = (\cdot, v) \\ 0 & \text{otherwise} \end{cases} . \tag{1}$$

The boundary operator can be used to define a Laplace operator on the edges of the graph, the so-called (Equivariant) Edge Laplacian (Schaub et al., 2021): $\boldsymbol{L}_{\mathrm{equ}} = \boldsymbol{B}_{\mathrm{equ}}^T \boldsymbol{B}_{\mathrm{equ}}$. This operator can be understood as message passing between edges that are incident to a shared node (see Section 4.3).

## 3 RELATED WORK

**Topological Models.** Methods grounded in Algebraic Topology often represent orientation-equivariant signals on undirected graphs relative to an arbitrary orientation (Schaub et al., 2021). Some of these approaches also utilize higher-order structures composed of edges that, however, often need to be handcrafted (Bunch et al., 2020; Giusti et al., 2022). HodgeGNN (Roddenberry & Segarra, 2019) and similar architectures (Park et al., 2023; Roddenberry et al., 2021) satisfy a more limited notion of orientation equivariance compared to our proposal. This results in major shortcomings of these models in practice: (i) They treat all input and output signals as orientation-equivariant and, therefore, can not model orientation-invariant edge signals appropriately. In the example of Figure 1, they do not predict the same orientation-invariant output under different orientations $\mathfrak{O}$ and $\hat{\mathfrak{O}}$ but instead induce sign flips as if they were orientation-equivariant signals leading to inconsistent predictions for different (arbitrary) orientations of the same topology. (ii) These approaches are equivariant regarding the orientation of *all* edges, whereas our desiderata relax this requirement to hold for *undirected* edges only. As depicted in Figure 2 , edge direction often alters the nature of the problem: In one scenario cars go against the direction of a one-way street while in the

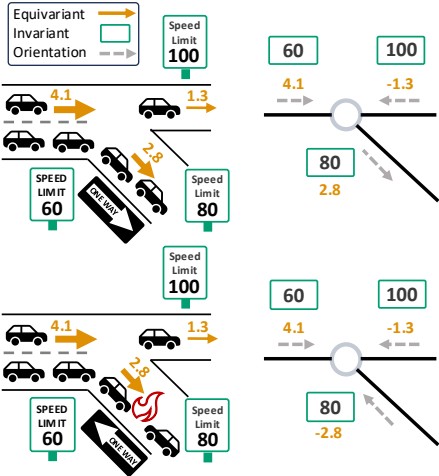

(a) Traffic Scenarios    (b) Representations

Figure 2: Two scenarios (top, bottom) that differ in the direction of one edge but model different situations (flame in bottom left). Their representations are indistinguishable for models that are orientation-equivariant for *directed* edges.

other they do not. For fully orientation equivariant models, both representations are indistinguishable: The edge orientation serves only as a representation basis and can not encode directionality. An exception among topological methods is concurrent work (Lecha et al., 2024) that implicitly distinguishes between directed and undirected edges by representing undirected edges with two directed yet antiparallel edges in a directed graph that is augmented with higher-order structures. However, they neither discuss the combination of equivariant and invariant features/targets nor is their model applicable to this setting without modifications.

**Flow Interpolation.** An orthogonal line of research studies flow interpolation problems (Ford & Fulkerson, 1956; Lippi et al., 2013). Even though both features and targets are technically orientation-equivariant, many methods do not approach this task from a topological perspective. Instead, they utilize specialized inductive biases such as flow conservation (Jia et al., 2019) or physics-informed constraints (Smith et al., 2022). da Silva et al. (2021) frame interpolation as a bi-level optimization problem where edge-level models serve as learned regularizers. Such approaches are limited to flow interpolation problems while our model is a general framework that can be applied to a broader range of edge-level tasks. As we find in Section 5.3, EIGN can learn the physical properties of a problem without explicitly encoding physics-informed inductive biases.

**Edge-Level GNNs.** Beyond flow-based problems, GNNs have been applied to (orientation-invariant) edge-level tasks as well. One family of approaches employs node-level GNNs and a successive readout function (Zhao et al., 2023). Such approaches have been particularly popular for link prediction (Zhou, 2021), where target edges are not present in the input data. Another paradigm is to apply node-level GNNs to the dual Line Graph (Jiang et al., 2019; Jo et al., 2021) of the topology. Only a few approaches rely on edge-specific methods (Zhang et al., 2020). Models that are not grounded in a topological framework lack appropriate inductive biases to model orientation-equivariant signals with inherent direction. These approaches can be categorized as treating both input and target as orientation-invariant signals. In the context of Figure 1, they would not represent orientation-equivariant signals relative to the respective orientation.

**GNNs for Directed Graphs.** Directed graphs have received a lot of attention for node-level problems. Many approaches discriminate between adjacent in-neighbors and out-neighbors (Li et al., 2016; Rossi et al., 2023). Also, spectral convolutions, that are based on the Node Laplacian, have been generalized to directed settings (Ma et al., 2019; Monti et al., 2018). From the different possible direction-aware Laplacians (Tong et al., 2020), our work takes inspiration from the Magnetic Node Laplacian (Zhang et al., 2021) which can be defined using a complex-valued boundary operator. It was recently utilized to compute direction-aware positional node encodings (Geisler et al., 2023). The Laplace operators of our approach generalize this concept to edge-level problems.

## 4 METHOD

### 4.1 DESIRABLE PROPERTIES FOR EDGE-LEVEL GNNS

At the core of our work stand novel desiderata which enforce a model to make consistent predictions for both orientation-equivariant and -invariant edge signals among different orientations. We restrict our novel constraints to *undirected edges* by requiring equivariance/invariance among direction-consistent orientations only. We additionally prove that EIGN is also equivariant with respect to edge permutations, which we defer to Appendix A.2.

**Definition 4.1** (Joint Orientation Equivariance). Let $\mathfrak{O}, \hat{\mathfrak{O}}$ be arbitrary *direction-consistent* orientations of edges on $\mathcal{G}$. We say that a mapping $f$ is jointly orientation-equivariant if for any orientation-equivariant input $\boldsymbol{X}_{\text{equ}} \in \mathbb{C}^{m \times d_{\text{equ}}}$ and any orientation-invariant input $\boldsymbol{X}_{\text{inv}} \in \mathbb{C}^{m \times d_{\text{inv}}}$:

$$\Delta_{\mathfrak{O}, \hat{\mathfrak{O}}} f(\boldsymbol{X}_{\text{equ}}, \boldsymbol{X}_{\text{inv}}, \mathcal{G}, \mathfrak{O}) = f(\Delta_{\mathfrak{O}, \hat{\mathfrak{O}}} \boldsymbol{X}_{\text{equ}}, \boldsymbol{X}_{\text{inv}}, \mathcal{G}, \hat{\mathfrak{O}}).$$

**Definition 4.2** (Joint Orientation Invariance). Let $\mathfrak{O}, \hat{\mathfrak{O}}$ be arbitrary *direction-consistent* orientations of edges on $\mathcal{G}$. We say that a mapping $g$ is jointly orientation-invariant if for any orientation-equivariant input $\boldsymbol{X}_{\text{equ}} \in \mathbb{C}^{m \times d_{\text{equ}}}$ and any orientation-invariant input $\boldsymbol{X}_{\text{inv}} \in \mathbb{C}^{m \times d_{\text{inv}}}$:

$$g(\boldsymbol{X}_{\text{equ}}, \boldsymbol{X}_{\text{inv}}, \mathcal{G}, \mathfrak{O}) = g(\Delta_{\mathfrak{O}, \hat{\mathfrak{O}}} \boldsymbol{X}_{\text{equ}}, \boldsymbol{X}_{\text{inv}}, \mathcal{G}, \hat{\mathfrak{O}}).$$

Both definitions ensure that a model predicts the same output signal when the orientation is changed from $\mathfrak{O}$ to $\hat{\mathfrak{O}}$: While Definition 4.1 ensures that the orientation-equivariant output is represented

relative to the new orientation $\hat{\mathfrak{O}}$, Definition 4.2 requires the same orientation-invariant predictions which are not relative to the new orientation. In both desiderata, the orientation-equivariant input $\boldsymbol{X}_{\text{equ}}$ needs to be represented relative to the respective orientation as well. Restricting these properties to direction-consistent orientations means that both properties only need to hold regarding the arbitrary orientation of undirected edges. This allows models to break both desiderata for the orientation of *directed edges*, which enables using their orientation to encode direction.

## 4.2 EIGN: AN ORIENTATION-EQUIVARIANT AND ORIENTATION-INVARIANT MODEL

We propose EIGN (Figure 3), a model that satisfies these desiderata. It consists of $L$ layers each of which takes orientation-equivariant and -invariant input edge signals $\boldsymbol{H}_{equ}^{(l-1)}$, $\boldsymbol{H}_{inv}^{(l-1)}$ and transforms them into outputs of the corresponding modality $\boldsymbol{H}_{\text{equ}}^{(l)}$ and $\boldsymbol{H}_{\text{inv}}^{(l)}$. Its message passing between edge signals is based on different graph shift operators (see "Convolutions" in Figure 3 and Section 4.3 for details) both within and between edge signal modalities. We set $\boldsymbol{H}_{\text{equ}}^{(0)} = \boldsymbol{X}_{\text{equ}}$, $\boldsymbol{H}_{\text{inv}}^{(0)} = \boldsymbol{X}_{\text{inv}}$ and $\boldsymbol{H}_{\text{equ}}^{(L)}$ and $\boldsymbol{H}_{\text{inv}}^{(L)}$ are the equivariant and invariant output signals, respectively. In the following, we denote with $\boldsymbol{W}_{(\cdot)}^{(l)}$ model parameters of appropriate dimensions, with $\sigma_{\text{equ}}$ an element-wise sign equivariant activation function, i.e. $\sigma_{\text{equ}}(-x) = -\sigma_{\text{equ}}(x)$, and with $\sigma_{\text{inv}}$ an arbitrary non-linearity. We detail the implementation of EIGN in Appendix D.3.

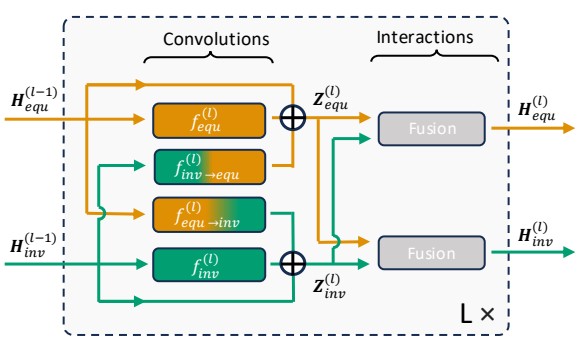

Figure 3: EIGN architecture: In each layer, message passing using novel Laplacians is performed within and between orientation-equivariant and orientation-invariant signals. The two aggregated modalities $\boldsymbol{Z}_{\text{equ}}^{(l)}$ and then $\boldsymbol{Z}_{\text{inv}}^{(l)}$ are then fused.

$$\boldsymbol{Z}_{\text{equ}}^{(l)} = \sigma_{\text{equ}}(f_{\text{equ}}^{(l)}(\boldsymbol{H}_{\text{equ}}^{(l-1)})\boldsymbol{W}_{\text{equ}\to\text{equ}}^{(l)} + f_{\text{inv}\to\text{equ}}^{(l)}(\boldsymbol{H}_{\text{inv}}^{(l-1)})\boldsymbol{W}_{\text{inv}\to\text{equ}}^{(l)} + \boldsymbol{H}_{\text{equ}}^{(l-1)}\boldsymbol{W}_{\text{equ}}^{(l)}). \tag{2}$$

$$\boldsymbol{Z}_{\text{inv}}^{(l)} = \sigma_{\text{inv}}(f_{\text{inv}}^{(l)}(\boldsymbol{H}_{\text{inv}}^{(l-1)})\boldsymbol{W}_{\text{inv}\to\text{inv}}^{(l)} + f_{\text{equ}\to\text{inv}}^{(l)}(\boldsymbol{H}_{\text{equ}}^{(l-1)})\boldsymbol{W}_{\text{equ}\to\text{inv}}^{(l)} + \boldsymbol{H}_{\text{inv}}^{(l-1)}\boldsymbol{W}_{\text{inv}}^{(l)}). \tag{3}$$

The intra-modality message passing schemes $f_{\text{equ}}^{(l)}$ and $f_{\text{inv}}^{(l)}$ update the orientation-equivariant and -invariant signal representation of an edge by aggregating orientation-equivariant and -invariant signals of adjacent edges. Similarly, the inter-modality schemes $f_{\text{inv}\to\text{equ}}^{(l)}$ and $f_{\text{equ}\to\text{inv}}^{(l)}$ aggregate messages from adjacent edges of one modality and transform it into the other. While this enables information exchange between directed and undirected edge signals, it restricts their interaction to local aggregates: The edge orientation-equivariant signal only depends on the average of adjacent orientation-invariant signals and vice versa. This makes modeling interactions between orientation-equivariant and –invariant edge signals of the same edge difficult. Therefore, EIGN uses a second fusion operation that does not use Laplacians (depicted as "Fusion" in Figure 3):

$$\boldsymbol{H}_{\text{equ}}^{(l)} = \sigma_{\text{equ}}(\boldsymbol{Z}_{\text{equ}}^{(l)}\boldsymbol{W}_{\text{F,equ}\to\text{equ}}^{(l)} \odot \boldsymbol{Z}_{\text{inv}}^{(l)}\boldsymbol{W}_{\text{F,inv}\to\text{equ}}^{(l)} + \boldsymbol{Z}_{\text{equ}}^{(l)}). \tag{4}$$

$$\boldsymbol{H}_{\text{inv}}^{(l)} = \sigma_{\text{inv}}(\boldsymbol{Z}_{\text{inv}}^{(l)}\boldsymbol{W}_{\text{F,inv}\to\text{inv}}^{(l)} \odot \text{abs}(\boldsymbol{Z}_{\text{equ}}^{(l)}\boldsymbol{W}_{\text{F,equ}\to\text{inv}}^{(l)}) + \boldsymbol{Z}_{\text{inv}}^{(l)}). \tag{5}$$

Here, $\odot$ and $\text{abs}$ denote element-wise multiplication and absolute value. In general, the fusion operation can be realized arbitrarily. We chose point-wise multiplication and absolute values as they are jointly orientation-equivariant / -invariant fusion operations respectively (see Appendix A).

EIGN models all possible types of interactions between edge signal modalities (Table 1): The Laplacian operators of Equations (2) and (3) describe interactions of orientation-equivariant and -invariant signals with local aggregates of both the same and different modality. The residual connections in Equations (2) and (3) and the fusion operation of

Table 1: Convolution and self- interaction operators for both input and output signal modalities.

| Output Input | Equ. | Inv. |
|---|---|---|
| Equ. | $\boldsymbol{L}_{\text{equ}}^{(q)}$ $\boldsymbol{W}_{\text{equ}}$ | $\boldsymbol{L}_{\text{equ}\to\text{inv}}^{(q)}$ $\boldsymbol{W}_{\text{F,equ}\to\text{inv}}$ |
| Inv. | $\boldsymbol{L}_{\text{inv}\to\text{equ}}^{(q)}$ $\boldsymbol{W}_{\text{F,inv}\to\text{equ}}$ | $\boldsymbol{L}_{\text{inv}}^{(q)}$ $\boldsymbol{W}_{\text{inv}}$ |

Equations (4) and (5) model interactions between orientation-equivariant and -invariant signals of the same edge. The design choices of the fusion operation in Equations (4) and (5) and the definition of the message passing schemes $f_{\text{equ}}^{(l)}$, $f_{\text{inv}}^{(l)}$, $f_{\text{inv}\to\text{equ}}^{(l)}$ and $f_{\text{equ}\to\text{inv}}^{(l)}$, which we detail next, enforce EIGN to conform to all desiderata proposed in Section 4.1.

### 4.3 ORIENTATION-EQUIVARIANT AND ORIENTATION-INVARIANT LAPLACIANS

**Equivariant and Invariant Edge Laplacians**. The Equivariant Edge Laplacian (see Section 2) arises from an (equivariant) boundary operator as $\boldsymbol{L}_{\text{equ}} = \boldsymbol{B}_{\text{equ}}^T \boldsymbol{B}_{\text{equ}}$. Its sparsity pattern corresponds to the adjacency matrix of the dual Line Graph (Jiang et al., 2019; Jo et al., 2021), i.e. edges are adjacent if they are incident to the same node. It is a jointly orientation-equivariant mapping (see Lemma 4.1) and therefore suitable to convole orientation-equivariant edge signals. Intuitively, $\boldsymbol{L}_{\text{equ}}$ performs message passing: A target edge $e \in \mathcal{E}$ aggregates the orientation-equivariant signals of adjacent edges. Additionally, it re-orients the signal of an adjacent edge $e' \in \mathcal{E}$ by multiplying with $-1$ if the orientations of $e$ and $e'$ misalign. The orientations of two edges misalign if they are consecutive, i.e. the endpoint of one is the starting point of the other (see Figure 4).

$$[\boldsymbol{L}_{\text{equ}}]_{e,e'} = \begin{cases} 2 & \text{if } e = e' \\ -1 & \text{if } \mathfrak{O}(e), \mathfrak{O}(e') \text{ consecutive} \\ 1 & \text{if } \mathfrak{O}(e), \mathfrak{O}(e') \text{ not consecutive, but } e, e' \text{ adjacent} \\ 0 & \text{otherwise} \end{cases}. \qquad (6)$$

We propose a novel Laplacian for orientation-invariant edge signals by using the orientation-independent unsigned boundary operator (Bodnar et al., 2021; Papillon et al., 2023):

$$[\boldsymbol{B}_{\text{inv}}]_{v,e} = \begin{cases} 1 & \text{if } v \in e \\ 0 & \text{otherwise} \end{cases}. \qquad (7)$$

It induces the Invariant Edge Laplacian $\boldsymbol{L}_{\text{inv}} = \boldsymbol{B}_{\text{inv}}^T \boldsymbol{B}_{\text{inv}}$ which has the same sparsity pattern as $\boldsymbol{L}_{\text{equ}}$. In particular, the Invariant Edge Laplacian can be obtained by taking the element-wise absolute value $\boldsymbol{L}_{\text{inv}} = \text{abs}(\boldsymbol{L}_{\text{equ}})$. Its message passing scheme is similar to $\boldsymbol{L}_{\text{equ}}$ as well: It, too, aggregates messages of adjacent edges but does not re-orient them if their orientations are consecutive. Thus, it is a jointly orientation-invariant mapping and suitable to convole orientation-invariant edge signals.

**Direction-aware Edge Laplacians.** Neither $\boldsymbol{L}_{\text{equ}}$ nor $\boldsymbol{L}_{\text{inv}}$ can distinguish directed from undirected edges as they do not explicitly model them differently. However, following a recent line of work on Magnetic Node Laplacians (Forman, 1993; Shubin, 1994; Colin de Verdière, 2013; Furutani et al., 2019; Geisler et al., 2023), we generalize both operators to make them direction-aware. Intuitively, the Magnetic Node Laplacian represents edge direction through complex phase shifts of magnitude $\pi q$ for some fixed hyperparamter $q \in \mathbb{R}$. It can be computed using a complex-valued boundary operator (Fanuel & Bardenet, 2024):

Figure 4: The Equivariant Magnetic Edge Laplacian $\boldsymbol{L}_{\text{equ}}^{(q)}$ induces a complex phase shift of $\pi q$ for signals of directed edges that are aggregated by the black undirected edge. Signals of misaligned edges are re-oriented.

$$\left[\boldsymbol{B}_{\text{equ}}^{(q)}\right]_{v,e} = \begin{cases} -\exp(i\pi q) & \text{if } e = (v, \cdot) \text{ and } e \in \mathcal{E}_{\text{D}} \\ \exp(-i\pi q) & \text{if } e = (\cdot, v) \text{ and } e \in \mathcal{E}_{\text{D}} \\ -1 & \text{if } \mathfrak{O}(e) = (v, \cdot) \text{ and } e \in \mathcal{E}_{\text{U}} \\ 1 & \text{if } \mathfrak{O}(e) = (\cdot, v) \text{ and } e \in \mathcal{E}_{\text{U}} \\ 0 & \text{otherwise} \end{cases}. \qquad (8)$$

This boundary operator $\boldsymbol{B}_{\text{equ}}^{(q)}$ extends $\boldsymbol{B}_{\text{equ}}$ by inducing complex phase shifts of $\pi q$ along directed edges. Consequently, $q = 0$ recovers the direction-agnostic boundary operator of Equation (1). Using the boundary operator $\boldsymbol{B}_{\text{equ}}^{(q)}$, we can define a direction-aware Equivariant Magnetic Edge Laplacian analogously to its direction-agnostic counterpart as $\boldsymbol{L}_{\text{equ}}^{(q)} = (\boldsymbol{B}_{\text{equ}}^{(q)})^H \boldsymbol{B}_{\text{equ}}^{(q)}$, with $(.)^H$ denoting the conjugate transposed . Its sparsity pattern is the same as $\boldsymbol{L}_{\text{equ}}$ and for undirected edges both operators coincide. Thus, its aggregation scheme is similar as well: The key difference is that $\boldsymbol{L}_{\text{equ}}^{(q)}$ applies a complex phase shift to the signals of *directed edges*

before aggregating them instead of just re-orienting them by flipping their sign. The direction of the phase shift is determined by how the edges relate to their shared incident node. The phases of ingoing edges will be shifted in a different direction than the phases of outgoing edges. Figure 4 depicts this mechanism for adjacent edges that are (i) undirected and aligned in orientation (ii) undirected and misaligned in orientation (iii) directed and aligned in orientation. $\boldsymbol{L}_{\text{equ}}^{(q)}$ is a jointly orientation-equivariant operator as per Definition 4.1 for the (arbitrary) orientation of undirected edges (even if they are adjacent to directed edges). It, however, specifically breaks orientation equivariance for *directed* edges by inducing complex phase shifts: Changing the direction of a directed edge flips the sign of the complex phase shift that is applied before aggregation (see Table 7).

Defining Laplace operators through boundary maps also enables a different interpretation of the induced message passing schemes. First, each node aggregates information from incident edges. Then each edge computes its representation from the information at its endpoints. In the case of equivariant signals, the node representations are analogous to potentials and the edge representations to the flow they induce. We utilize this through learnable node feature transformations and realize the message passing in Equations (2) and (3) as $f_{(\cdot)}^{(l)}(\boldsymbol{X}) = \boldsymbol{B}_{(\cdot)}^{H} h_{(\cdot)}^{(l)}(\boldsymbol{B}_{(\cdot)}\boldsymbol{X})$ instead of directly using the corresponding Laplacian $(\boldsymbol{B}_{(\cdot)})^{H}\boldsymbol{B}_{(\cdot)}$ as a graph shift operator.

**Lemma 4.1.** *The Magnetic Equivariant Edge Laplacian implies a jointly orientation-equivariant mapping $f(\boldsymbol{X}_{equ}, \boldsymbol{X}_{inv}, \mathcal{G}, \mathfrak{D}) = (\boldsymbol{B}_{equ}^{(q)})^{H} h(\boldsymbol{B}_{equ}^{(q)}\boldsymbol{X}_{equ})$.*

Analogously, we use complex phase shifts to encode direction into the boundary operator $\boldsymbol{B}_{\text{inv}}$ (Equation (7)) that induces the Laplacian $\boldsymbol{L}_{\text{inv}}$ for orientation-invariant signals:

$$\left[\boldsymbol{B}_{\text{inv}}^{(q)}\right]_{v,e} = \begin{cases} \exp(i\pi q) & \text{if } e = (v, \cdot) \text{ and } e \in \mathcal{E}_{\text{D}} \\ \exp(-i\pi q) & \text{if } e = (\cdot, v) \text{ and } e \in \mathcal{E}_{\text{D}} \\ 1 & \text{if } v \in e \text{ and } e \in \mathcal{E}_{\text{U}} \\ 0 & \text{otherwise} \end{cases}. \tag{9}$$

It induces a Magnetic Invariant Edge Laplacian $\boldsymbol{L}_{\text{inv}}^{(q)} = (\boldsymbol{B}_{\text{inv}}^{(q)})^{H}\boldsymbol{B}_{\text{inv}}^{(q)}$ that performs message passing analogous to the Magnetic Equivariant Edge Laplacian. Like its direction-agnostic counterpart $\boldsymbol{L}_{\text{inv}}$, it is defined independently of orientation and does not re-orient its inputs. Instead, it generalizes $\boldsymbol{L}_{\text{inv}}$ by only encoding edge direction through the direction of a complex phase shift similar to $\boldsymbol{L}_{\text{equ}}^{(q)}$. It is a jointly orientation-invariant mapping as per Definition 4.2.

**Lemma 4.2.** *The Magnetic Invariant Edge Laplacian implies a jointly orientation-invariant mapping $g(\boldsymbol{X}_{equ}, \boldsymbol{X}_{inv}, \mathcal{G}, \mathfrak{D}) = (\boldsymbol{B}_{inv}^{(q)})^{H} h(\boldsymbol{B}_{inv}^{(q)}\boldsymbol{X}_{inv})$.*

**Fusing Invariant and Equivariant Signals.** Both $\boldsymbol{L}_{\text{equ}}^{(q)}$ and $\boldsymbol{L}_{\text{inv}}^{(q)}$ are convolutions within edge signals of the same modality, i.e. orientation-equivariant or orientation-invariant signals. We enable information exchange between both by combining the boundary operators of Equations (8) and (9). First, we define a Laplacian to transform orientation-equivariant edge signals into orientation-invariant edge signals as $\boldsymbol{L}_{\text{equ}\rightarrow\text{inv}}^{(q)} = (\boldsymbol{B}_{\text{inv}}^{(q)})^{H}\boldsymbol{B}_{\text{equ}}^{(q)}$. This operator allows modeling orientation-invariant outputs even if no orientation-invariant inputs are available. Since it is constructed from direction-aware boundary operators it, too, induces complex phase shifts to encode directed edges.

Analogously, a fusion operator that transforms orientation-invariant edge signals into orientation-equivariant edge signals can be constructed as $\boldsymbol{L}_{\text{inv}\rightarrow\text{equ}}^{(q)} = (\boldsymbol{B}_{\text{equ}})^{H}\boldsymbol{B}_{\text{inv}}$. It transforms orientation-invariant inputs into an orientation-equivariant edge signal. Again, directed edges are encoded with complex phase shifts. The outputs of both operators satisfy joint orientation equivariance/invariance (Definitions 4.1 and 4.2) for undirected edges for which no complex phase shift is applied.

**Lemma 4.3.** *The Invariant and Equivariant Fusion Magentic Edge Laplacians implies a jointly orientation-invariant and jointly orientation-equivariant mapping $g(\boldsymbol{X}_{equ}, \boldsymbol{X}_{inv}, \mathcal{G}, \mathfrak{D}) = (\boldsymbol{B}_{inv}^{(q)})^{H} h(\boldsymbol{B}_{equ}^{(q)}\boldsymbol{X}_{equ})$ and a $f(\boldsymbol{X}_{equ}, \boldsymbol{X}_{inv}, \mathcal{G}, \mathfrak{D}) = (\boldsymbol{B}_{equ}^{(q)})^{H} h(\boldsymbol{B}_{inv}^{(q)}\boldsymbol{X}_{inv})$ respectively.*

Since EIGN is composed of Laplacians that are jointly orientation-equivariant/-invariant mappings respectively, it satisfies the desiderata for edge-level GNNs stated in Section 4.1. The proof follows Lemmata 4.1 to 4.3 and is supplied in Appendix A.

**Theorem 4.1.** *(i) $\boldsymbol{H}_{equ}^{(L)}$ is a jointly orientation-equivariant mapping. (ii) $\boldsymbol{H}_{inv}^{(L)}$ is a jointly orientation-invariant mapping. (iii) Both $\boldsymbol{H}_{equ}^{(L)}$ and $\boldsymbol{H}_{inv}^{(L)}$ are permutation equivariant mappings.*

Table 2: Modelling capabilities of all architectures. "-" denotes that the modality is modeled without satisfying orientation invariance/equivariance.

| Model | Edges | | Features | | Targets | |
|---|---|---|---|---|---|---|
| | Dir. | Undir. | Equ. | Inv. | Equ. | Inv. |
| MLP | ✗ | ✗ | ✗ | - | ✗ | - |
| LINEGRAPH | ✗ | ✓ | ✗ | ✓ | ✗ | ✓ |
| HODGEGNN | ✗ | ✓ | ✓ | ✗ | ✓ | ✗ |
| HODGE+INV | ✗ | ✓ | ✓ | - | ✓ | ✗ |
| HODGE+DIR | ✓ | ✗ | - | ✗ | - | ✗ |
| LINE-MAGNET | ✓ | ✓ | - | ✓ | - | ✓ |
| Dir-GNN* | ✓ | ✓ | ✓ | ✓ | ✓ | ✓ |
| EIGN* | ✓ | ✓ | ✓ | ✓ | ✓ | ✓ |

Table 3: Datasets in terms of edge direction and input/target feature modality.

| Dataset | Edges | | Features | | Targets |
|---|---|---|---|---|---|
| | Dir. | Undir. | Equ. | Inv. | |
| RW Comp | ✓ | | | ✓ | Inv. |
| LD Cycles | ✓ | ✓ | | | Inv. |
| Tri-Flow | ✓ | ✓ | ✓ | ✓ | Equ. |
| Anaheim | ✓ | ✓ | (✓) | ✓ | Equ. |
| Barcelona | ✓ | ✓ | (✓) | ✓ | Equ. |
| Chicago | | ✓ | (✓) | ✓ | Equ. |
| Winnipeg | ✓ | ✓ | (✓) | ✓ | Equ. |
| Circuits | ✓ | ✓ | ✓ | ✓ | Equ. |

# 5  EXPERIMENTS

We showcase the efficacy of EIGN on three synthetic problems and three tasks on five real-world datasets. We devise the synthetic tasks to require proper handling of directionality as well as orientation-equivariant and orientation-invariant information. We categorize all datasets in terms of whether the graphs are (partially) directed and if features and/or targets are orientation-equivariant or orientation-invariant in Table 3 (see details in Appendix D).

**Baselines.** We compare EIGN to five baselines: (i) An MLP. (ii) LINEGRAPHGNN, a node-level spectral GNN applied to the line graph of the problem (Bandyopadhyay et al., 2020)). (iii) HODGEGNN (Roddenberry et al., 2021; Park et al., 2023), based on the Edge Laplacian $L_{equ}$ (see Section 2): It assumes inputs to be orientation-equivariant and edges to be undirected. (iv) HODGE+INV as a variant of HODGE+INV that models orientation-invariant features as orientation-equivariant. (v) HODGE+DIR, a variant of HODGEGNN that breaks orientation equivariance, and, thus, treats all edges as directed. We also adapt two node-level GNNs for directed graphs (see Appendix D.3): (vi) LINE-MAGNET, a graph transformer similar to Geisler et al. (2023) and (vii) DIR-GNN, an edge-level GNN that represents directed edges through separate message-passing operations (Rossi et al., 2023; Battaglia et al., 2018). As depicted in Table 2, EIGN is applicable in every possible scenario. While Dir-GNN can be adapted to all modalities using our proposed Laplacians, this comes with significant drawbacks (see Appendix D.3). To mitigate side-effects from the cyclical nature of complex phase shifts, we choose $q = 1/m$ (see Appendix E).

## 5.1  SYNTHETIC TASKS

**Random Walk Completion (RW Comp).** The first synthetic problem we devise is completing random walks on a directed graph. We input the (orientation-invariant) transition probabilities and a subset of the edges that were traversed. The task is to classify if an edge is part of the random walk (details in Appendix D.1). Therefore, this problem tests if a model can distinguish different edge directions.

**Longest Directed Cycle Prediction (LD Cycles).** We increase the problem difficulty by including both directed and undirected edges: We generate different graphs that contain cycles entirely composed of each respective edge type. The task is to predict which edges belong to the largest cycle of only directed edges (details in

Table 4: Average performance of different models on synthetic tasks (**best** and **runner-up**). EIGN is particularly effective on the hard Tri-Flow problem with interactions between orientation-equivariant and orientation-invariant inputs.

| Model | RW Comp | LD Cycles | Tri-Flow |
|---|---|---|---|
| | AUC-ROC(↑) | AUC-ROC(↑) | RMSE(↓) |
| MLP | 0.720 | 0.500 | 0.547 |
| LINEGRAPH | 0.758 | 0.683 | 0.497 |
| HODGEGNN | 0.500 | 0.500 | 0.458 |
| HODGE+INV | 0.811 | 0.754 | **0.293** |
| HODGE+DIR | **0.819** | **0.799** | **0.293** |
| LINE-MAGNET | 0.729 | 0.502 | 0.542 |
| Dir-GNN | 0.757 | 0.768 | 0.453 |
| EIGN | **0.864** | **0.996** | **0.022** |

Appendix D.1). Since there are no additional inputs, this task tests a model's ability to distinguish directed and undirected edges.

**Triangle Flow Orientation (Tri-Flow).** This is the most challenging synthetic task as it requires combining orientation-equivariant and orientation-invariant features in a partially directed graph. We create multiple graphs containing disjoint triangles and provide an (orientation-equivariant) flow input. The task is to reorient the flow such that for triangles satisfying certain conditions there is no excess flux. These constraints are based on direction and an orientation-invariant attribute assigned to every edge, thus introducing a relationship between direction and both edge feature modalities.

**Results.** Table 4 shows that on all three synthetic tasks, EIGN is the superior architecture. Its high efficacy on the RW Comp and LD Cycles problems confirm its merits in modeling edge direction: It distinguishes between edges of different directions as well as undirected edges. As it satisfies our desiderata, it can model the relationship between orientation-equivariant and orientation-invariant features and relate them to edge direction. This is reflected in its impressive performance on the Tri-Flow task while the direction-aware HODGEGNN as well as HODGE+DIR, which also combines both orientation-equivariant and -invariant inputs, struggle to achieve comparable results.

## 5.2 REAL-WORLD PROBLEMS

**Datasets.** We also apply EIGN to real-world traffic networks and electrical circuits. (i) We select four transportation networks (Stabler et al., 2016) (Anaheim, Barcelona, Chicago, Winnipeg) where the targets are the best-known flow solutions in terms of lowest Average Excess Cost (Boyce et al., 2004). While these datasets only contain orientation-invariant features, two tasks use

Table 5: Average RMSE (↓) of different models for the simulation task on real-world datasets (**best** and **runner-up**). EIGN achieves substantial improvements over all baselines.

| Model | Anaheim | Barcelona | Chicago | Winnipeg | Circuits |
|-------|---------|-----------|---------|----------|----------|
| MLP | 0.105 | 0.149 | 0.109 | 0.167 | 1.030 |
| LINEGRAPH | 0.101 | 0.149 | 0.109 | 0.164 | 1.037 |
| HODGEGNN | 0.280 | 0.170 | 0.107 | 0.173 | 1.016 |
| HODGE+INV | 0.098 | 0.146 | 0.108 | 0.151 | 0.828 |
| HODGE+DIR | **0.091** | **0.144** | 0.109 | **0.132** | **0.760** |
| LINE-MAGNET | 0.119 | 0.151 | **0.105** | 0.170 | 1.027 |
| Dir-GNN | 0.278 | 0.170 | 0.106 | 0.173 | 1.029 |
| EIGN | **0.090** | **0.133** | **0.078** | **0.101** | **0.696** |

information from the orientation-equivariant targets as additional inputs. (ii) We generate different electrical circuits consisting of resistances, diodes, and one power outlet with a given voltage. We then simulate currents and voltages using LTSpice (Asadi, 2022). Each circuit has orientation-equivariant (voltage at the source) and orientation-invariant features (resistance, component type), while the target current is also orientation-equivariant (details in Appendix D.1 ).

**Tasks.** For each dataset, we study three problems of increasing difficulty for each of which the edge flow needs to be predicted: (i) *Denoising*: We noise the target flow and provide it as an (additional) equivariant input. (ii) *Interpolation*: We supply the target flow on a subset of edges as an (additional) equivariant input. (iii) *Simulation*: The target flow is to be predicted without any additional inputs.

**Results.** We report the RMSE (↓) of all models for the simulation task in Table 5 and defer results for the easier denoising and interpolation problems to Appendix E. EIGN substantially

Table 6: Ablation of different components of EIGN on synthetic tasks and simulation on real data (**best** and **runner-up**). We omit (i) direction-awarenes ($q = 0$), (ii) the fusion operation of Equations (4) and (5), (iii) the fusion operators $f_{\text{equ}\rightarrow\text{inv}}$ and $f_{\text{inv}\rightarrow\text{equ}}$, and (iv) the node embedding $h$.

| | Dataset | EIGN w/o Direction | EIGN No Fusion | EIGN No Fusion-Conv. | EIGN No $h$ | EIGN |
|---|---------|-----------|---------|--------------|-----------|------|
| AUC ↑ | RW Comp | 0.762 | 0.853 | 0.845 | **0.862** | **0.864** |
| | LD Cycles | 0.689 | **0.987** | 0.926 | **0.996** | **0.996** |
| RMSE ↓ | Tri-Flow | 0.362 | 0.088 | 0.074 | **0.034** | **0.022** |
| | Anaheim | 0.289 | **0.097** | 0.283 | 0.099 | **0.090** |
| | Barcelona | 0.172 | **0.139** | 0.177 | 0.163 | **0.133** |
| | Chicago | **0.079** | 0.093 | 0.110 | 0.082 | **0.078** |
| | Winnipeg | **0.132** | 0.170 | 0.175 | 0.138 | **0.101** |
| | Circuits | 0.957 | 0.974 | 0.727 | **0.707** | **0.696** |

improves over all baselines in interpolation. On the challenging simulation task, EIGN shows the merit of treating orientation-equivariant and orientation-invariant signals in a principled way.

## 5.3 ABLATIONS

**Fusion Operators.** In Section 4.3, we introduce Laplacian operators to fuse orientation-equivariant and orientation-invariant features by transforming a signal from one modality into the other. This is particularly useful as it allows EIGN to output orientation-equivariant signals in the absence of orientation-equivariant inputs and vice versa. Omitting them and only relying on Equations (4) and (5) for interaction between the two modalities is problematic for equivariant signals:

The only choice for an equivariant input signal in the absence of such features is $0$. Intuitively, this is because the sign of an orientation-equivariant input indicates its direction relative to the chosen orientation. However, the "absence" of an input has no inherent direction. Setting the equivariant input to zero while at the same time omitting the $f_{\text{inv}\to\text{equ}}$ term in Equation (2) results embeddings $\boldsymbol{Z}_{\text{equ}}^{(l)} = \boldsymbol{0}$ for undirected edges (see Appendix A.4), and, consequently, $\boldsymbol{H}^{(l)} = \boldsymbol{0}$ for Equation (4) as well. Therefore, omitting the Laplacian fusion operation from Equation (2) prohibits the model from learning any non-zero orientation-equivariant output in the absence of orientation-equivariant inputs. Table 6 confirms that omitting these fusion operators results in considerably worse performance in simulation tasks where no orientation-equivariant features are available.

At the same time, omitting the fusion operation of Equations (4) and (5) and only relying on the Laplacian fusion operators $f_{\text{equ}\to\text{inv}}$, $f_{\text{inv}\to\text{equ}}$ restricts the information exchange between orientation-equivariant and orientation-invariant features to only exchanging aggregated information through the convolutions $f_{\text{equ}\to\text{inv}}$ and $f_{\text{inv}\to\text{equ}}$. Therefore, it is more difficult to learn direct interactions between the signal modalities of *the same* edge. Both on Tri-Flow and real data, where these interactions are relevant, omitting the operators of Equations (4) and (5) leads to worse performance, see Table 6.

**Direction-Awareness.** We also study the impact of using complex-valued Laplace operators to make EIGN direction-aware by ablating a direction-agnostic version of EIGN in Table 6 ($q = 0$). First, the RW Comp and LD Cycles tasks that test direction-awareness benefit greatly from encoding direction with complex phase shifts. Second, many real-world datasets contain directed edges that constrain flows. Accurately modeling the system requires distinguishing between directed and undirected edges which is reflected in the superior performance of the direction-aware EIGN. Finally, Figure 5 shows that the direction-aware EIGN yields more plausible predictions than the direction-agnostic variant for one-way roads. It learns the problem constraints without encoding them into the architecture explicitly.

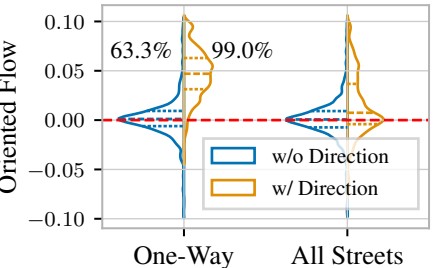

Figure 5: Distribution of simulated traffic flow on Anaheim for EIGN and a direction-agnostic variant. Direction-awareness makes predictions that obey flow constraints of one-way streets more likely.

## 6 LIMITATIONS

We address lack of benchmarks for edge-level tasks by designing three synthetic tasks and propose a novel task involving real-world electric circuits. Even though EIGN is grounded in Algebraic Topology, we limit our study to edge-level problems and do not model higher-order structures. W use Magnetic Laplacians to encode edge direction but other frameworks satisfying our formal desiderata may be effective as well in practice. Lastly, since our Laplacians are *normal matrices*, they enable global propagation similar to Geisler et al. (2024) to extend upon the local scheme used in our work.

## 7 CONCLUSION

We propose EIGN, a framework for edge-level problems that allows modeling orientation-equivariant and orientation-invariant features and can encode edge direction. It relies on novel graph shift operators that provably preserve novel notions of joint orientation-equivariance and -invariance for undirected edges while also being sensitive to flipping directed edges. On a benchmark of synthetic tasks and real-world flow modeling problems from two domains, we show the high efficacy of the inductive biases encoded by EIGN.

ETHICS STATEMENT

We acknowledge that we thoroughly read and adhere to the code of ethics. Since our work can be categorized as foundational research, we do not see any immediate implications beyond the risk of advancing Machine Learning, in general. We, nonetheless, encourage readers and practitioners building on our work to keep in mind the potential risks in the context of reliability, interpretability, fairness, and privacy for which we do explicitly account.

REPRODUCABILITY STATEMENT

We detail assumptions and proofs for all claims made in our work clearly in Appendix A. Furthermore, we describe in detail how the data and models are tuned in Appendix D. Additionally, we provide our code and the optimal hyperparameter configurations we found in the supplementary material.

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

## A  PROOFS

### A.1  JOINT ORIENTATION EQUIVARIANCE AND JOINT ORIENTATION INVARIANCE

**Proposition A.1.** *Let $\mathfrak{O}, \hat{\mathfrak{O}}$ be two direction-consistent orientations of edges on $\mathcal{G}$. Let $\boldsymbol{X}_{equ} \in \mathbb{C}^{m \times d}$. Then:*

$$\boldsymbol{B}_{equ}^{(q)}(\mathcal{G}, \mathfrak{O}) \Delta_{\mathfrak{O}, \hat{\mathfrak{O}}} = \boldsymbol{B}_{equ}^{(q)}(\mathcal{G}, \hat{\mathfrak{O}})$$

*Proof.* By assumption, for any $e_{\mathrm{D}} \in \mathcal{E}_{\mathrm{D}}$ we have that $\hat{\mathfrak{O}}(e_{\mathrm{D}}) = \mathfrak{O}(e_{\mathrm{D}}) = \Delta_{\mathfrak{O}, \hat{\mathfrak{O}}} \mathfrak{O}(e_{\mathrm{D}})$. Now consider the case $e_{\mathrm{U}} \in \mathcal{E}_{\mathrm{U}}$. If $\mathfrak{O}(e_{\mathrm{U}}) = \hat{\mathfrak{O}}(e_{\mathrm{U}})$, then $\boldsymbol{B}_{\mathrm{equ}}^{(q)}(\mathcal{G}, \mathfrak{O})_{v, e_{\mathrm{U}}} = \boldsymbol{B}_{\mathrm{equ}}^{(q)}(\mathcal{G}, \hat{\mathfrak{O}})_{v, e_{\mathrm{U}}}$ for all $v \in \mathcal{V}$. Similarly, if $\mathfrak{O}(e_{\mathrm{U}}) \neq \hat{\mathfrak{O}}(e_{\mathrm{U}})$, then $[\boldsymbol{B}_{\mathrm{equ}}^{(q)}(\mathcal{G}, \mathfrak{O})]_{v, e} = -[\boldsymbol{B}_{\mathrm{equ}}^{(q)}(\mathcal{G}, \hat{\mathfrak{O}})]_{v, e_{\mathrm{U}}}$ for all $v \in \mathcal{V}$. □

**Lemma 4.1.** *The Magnetic Equivariant Edge Laplacian implies a jointly orientation-equivariant mapping $f(\boldsymbol{X}_{equ}, \boldsymbol{X}_{inv}, \mathcal{G}, \mathfrak{O}) = (\boldsymbol{B}_{equ}^{(q)})^H h(\boldsymbol{B}_{equ}^{(q)} \boldsymbol{X}_{equ})$.*

*Proof.* Let $\mathfrak{O}, \hat{\mathfrak{O}}$ be two direction-consistent orientations of edges on $\mathcal{G}$. Let $\boldsymbol{X}_{\mathrm{equ}} \in \mathbb{C}^{m \times d_{\mathrm{equ}}}$ and $\boldsymbol{X}_{\mathrm{inv}} \in \mathbb{C}^{m \times d_{\mathrm{inv}}}$. Then:

$$\begin{aligned}
f(\Delta_{\mathfrak{O}, \hat{\mathfrak{O}}} \boldsymbol{X}_{\mathrm{equ}}, \boldsymbol{X}_{\mathrm{inv}}, \mathcal{G}, \hat{\mathfrak{O}}) &= (\boldsymbol{B}_{\mathrm{equ}}^{(q)}(\mathcal{G}, \hat{\mathfrak{O}}))^H h(\boldsymbol{B}_{\mathrm{equ}}^{(q)}(\mathcal{G}, \hat{\mathfrak{O}}) \Delta_{\mathfrak{O}, \hat{\mathfrak{O}}} \boldsymbol{X}_{\mathrm{equ}}) \\
&= (\boldsymbol{B}_{\mathrm{equ}}^{(q)}(\mathcal{G}, \mathfrak{O}) \Delta_{\mathfrak{O}, \hat{\mathfrak{O}}})^H h(\boldsymbol{B}_{\mathrm{equ}}^{(q)}(\mathcal{G}, \mathfrak{O}) \Delta_{\mathfrak{O}, \hat{\mathfrak{O}}} \Delta_{\mathfrak{O}, \hat{\mathfrak{O}}} \boldsymbol{X}_{\mathrm{equ}}) \\
&= (\Delta_{\mathfrak{O}, \hat{\mathfrak{O}}})^H (\boldsymbol{B}_{\mathrm{equ}}^{(q)}(\mathcal{G}, \mathfrak{O}))^H h(\boldsymbol{B}_{\mathrm{equ}}^{(q)}(\mathcal{G}, \mathfrak{O}) \boldsymbol{X}_{\mathrm{equ}}) \\
&= \Delta_{\mathfrak{O}, \hat{\mathfrak{O}}} f(\boldsymbol{X}_{\mathrm{equ}}, \boldsymbol{X}_{\mathrm{inv}}, \mathcal{G}, \mathfrak{O})
\end{aligned}$$

Here, we used Proposition A.1 and the facts that $\Delta_{\mathfrak{O}, \hat{\mathfrak{O}}} \Delta_{\mathfrak{O}, \hat{\mathfrak{O}}} = \boldsymbol{I}_m$ by definition and $\Delta_{\mathfrak{O}, \hat{\mathfrak{O}}} = (\Delta_{\mathfrak{O}, \hat{\mathfrak{O}}})^H$ as it is a diagonal real matrix. The proof holds for arbitrary node feature transformations $h$.

□

**Proposition A.2.** *Let $\mathfrak{O}, \hat{\mathfrak{O}}$ be two direction-consistent orientations of edges on $\mathcal{G}$. Let $\boldsymbol{X}_{equ} \in \mathbb{C}^{m \times d}$. Then:*

$$\boldsymbol{B}_{inv}^{(q)}(\mathcal{G}, \mathfrak{O}) = \boldsymbol{B}_{inv}^{(q)}(\mathcal{G}, \hat{\mathfrak{O}})$$

*Proof.* By assumption, for any $e_{\mathrm{D}} \in \mathcal{E}_{\mathrm{D}}$ we have that $\mathfrak{O}(\hat{e}_{\mathrm{D}}) = \mathfrak{O}(e_{\mathrm{D}}) = \Delta_{\mathfrak{O}, \hat{\mathfrak{O}}} \mathfrak{O}(e_{\mathrm{D}})$ and consequentially, $[\boldsymbol{B}_{\mathrm{inv}}^{(q)}(\mathcal{G}, \mathfrak{O})]_{v, e_{\mathrm{D}}} = [\boldsymbol{B}_{\mathrm{inv}}^{(q)}(\mathcal{G}, \hat{\mathfrak{O}})]_{v, e_{\mathrm{D}}}$ for all $v \in \mathcal{V}$. For $e_{\mathrm{U}} \in \mathcal{E}_{\mathrm{U}}$, the claim follows directly from Equation (9). □

**Lemma 4.2.** *The Magnetic Invariant Edge Laplacian implies a jointly orientation-invariant mapping $g(\boldsymbol{X}_{equ}, \boldsymbol{X}_{inv}, \mathcal{G}, \mathfrak{O}) = (\boldsymbol{B}_{inv}^{(q)})^H h(\boldsymbol{B}_{inv}^{(q)} \boldsymbol{X}_{inv})$.*

*Proof.* Let $\mathfrak{O}, \hat{\mathfrak{O}}$ be two direction-consistent orientations of edges on $\mathcal{G}$. Let $\boldsymbol{X}_{\mathrm{equ}} \in \mathbb{C}^{m \times d_{\mathrm{equ}}}$ and $\boldsymbol{X}_{\mathrm{inv}} \in \mathbb{C}^{m \times d_{\mathrm{inv}}}$. Then:

$$\begin{aligned}
g(\boldsymbol{X}_{\mathrm{equ}}, \boldsymbol{X}_{\mathrm{inv}}, \mathcal{G}, \hat{\mathfrak{O}}) &= (\boldsymbol{B}_{\mathrm{inv}}^{(q)}(\mathcal{G}, \hat{\mathfrak{O}}))^H h(\boldsymbol{B}_{\mathrm{inv}}^{(q)}(\mathcal{G}, \hat{\mathfrak{O}}) \boldsymbol{X}_{\mathrm{inv}}) \\
&= (\boldsymbol{B}_{\mathrm{inv}}^{(q)}(\mathcal{G}, \mathfrak{O}))^H h(\boldsymbol{B}_{\mathrm{inv}}^{(q)}(\mathcal{G}, \mathfrak{O}) \boldsymbol{X}_{\mathrm{inv}}) \\
&= g(\boldsymbol{X}_{\mathrm{equ}}, \boldsymbol{X}_{\mathrm{inv}}, \mathcal{G}, \mathfrak{O})
\end{aligned}$$

Here, we directly applied Proposition A.2. The proof holds for arbitrary node feature transformations $h$. □

**Lemma 4.3.** *The Invariant and Equivariant Fusion Magentic Edge Laplacians implies a jointly orientation-invariant and jointly orientation-equivariant mapping $g(\boldsymbol{X}_{equ}, \boldsymbol{X}_{inv}, \mathcal{G}, \mathfrak{O}) = (\boldsymbol{B}_{inv}^{(q)})^H h(\boldsymbol{B}_{equ}^{(q)} \boldsymbol{X}_{equ})$ and a $f(\boldsymbol{X}_{equ}, \boldsymbol{X}_{inv}, \mathcal{G}, \mathfrak{O}) = (\boldsymbol{B}_{equ}^{(q)})^H h(\boldsymbol{B}_{inv}^{(q)} \boldsymbol{X}_{inv})$ respectively.*

*Proof.* Let $\mathfrak{O}, \hat{\mathfrak{O}}$ be two direction-consistent orientations of edges on $\mathcal{G}$. Let $\boldsymbol{X}_{\text{equ}} \in \mathbb{C}^{m \times d_{\text{equ}}}$ and $\boldsymbol{X}_{\text{inv}} \in \mathbb{C}^{m \times d_{\text{inv}}}$. Then:

$$
\begin{aligned}
g(\Delta_{\mathfrak{O}, \hat{\mathfrak{O}}} \boldsymbol{X}_{\text{equ}}, \boldsymbol{X}_{\text{inv}}, \mathcal{G}, \hat{\mathfrak{O}}) &= (\boldsymbol{B}_{\text{inv}}^{(q)}(\mathcal{G}, \hat{\mathfrak{O}}))^H h(\boldsymbol{B}_{\text{equ}}^{(q)}(\mathcal{G}, \hat{\mathfrak{O}}) \Delta_{\mathfrak{O}, \hat{\mathfrak{O}}} \boldsymbol{X}_{\text{equ}}) \\
&= (\boldsymbol{B}_{\text{inv}}^{(q)}(\mathcal{G}, \mathfrak{O}))^H h(\boldsymbol{B}_{\text{equ}}^{(q)}(\mathcal{G}, \mathfrak{O}) \Delta_{\mathfrak{O}, \hat{\mathfrak{O}}} \Delta_{\mathfrak{O}, \hat{\mathfrak{O}}} \boldsymbol{X}_{\text{equ}}) \\
&= (\boldsymbol{B}_{\text{inv}}^{(q)}(\mathcal{G}, \mathfrak{O}))^H h(\boldsymbol{B}_{\text{equ}}^{(q)}(\mathcal{G}, \mathfrak{O}) \boldsymbol{X}_{\text{equ}}) \\
&= g(\boldsymbol{X}_{\text{equ}}, \boldsymbol{X}_{\text{inv}}, \mathcal{G}, \mathfrak{O})
\end{aligned}
$$

Here, we used Propositions A.1 and A.2 and the fact that $\Delta_{\mathfrak{O}, \hat{\mathfrak{O}}} \Delta_{\mathfrak{O}, \hat{\mathfrak{O}}} = \boldsymbol{I}_m$.

Let $\mathfrak{O}, \hat{\mathfrak{O}}$ be two direction-consistent orientations of edges on $\mathcal{G}$. Let $\boldsymbol{X}_{\text{equ}} \in \mathbb{C}^{m \times d_{\text{equ}}}$ and $\boldsymbol{X}_{\text{inv}} \in \mathbb{C}^{m \times d_{\text{inv}}}$. Then:

$$
\begin{aligned}
f(\Delta_{\mathfrak{O}, \hat{\mathfrak{O}}} \boldsymbol{X}_{\text{equ}}, \boldsymbol{X}_{\text{inv}}, \mathcal{G}, \hat{\mathfrak{O}}) &= (\boldsymbol{B}_{\text{equ}}^{(q)}(\mathcal{G}, \hat{\mathfrak{O}}))^H h(\boldsymbol{B}_{\text{inv}}^{(q)}(\mathcal{G}, \hat{\mathfrak{O}}) \boldsymbol{X}_{\text{inv}}) \\
&= (\boldsymbol{B}_{\text{equ}}^{(q)}(\mathcal{G}, \mathfrak{O}) \Delta_{\mathfrak{O}, \hat{\mathfrak{O}}})^H h(\boldsymbol{B}_{\text{inv}}^{(q)}(\mathcal{G}, \mathfrak{O}) \boldsymbol{X}_{\text{inv}}) \\
&= (\Delta_{\mathfrak{O}, \hat{\mathfrak{O}}})^H (\boldsymbol{B}_{\text{equ}}^{(q)}(\mathcal{G}, \mathfrak{O}))^H h(\boldsymbol{B}_{\text{inv}}^{(q)}(\mathcal{G}, \mathfrak{O}) \boldsymbol{X}_{\text{inv}}) \\
&= \Delta_{\mathfrak{O}, \hat{\mathfrak{O}}} f(\boldsymbol{X}_{\text{equ}}, \boldsymbol{X}_{\text{inv}}, \mathcal{G}, \mathfrak{O})
\end{aligned}
$$

Here, we used Propositions A.1 and A.2 and the fact that $(\Delta_{\mathfrak{O}, \hat{\mathfrak{O}}})^H = \Delta_{\mathfrak{O}, \hat{\mathfrak{O}}}$ as it is a diagonal real matrix. $\qquad \square$

We now prove that the composition of orientation-equivariant / invariant functions preserves orientation equivariance / invariance, which allows us to prove Theorem 4.1 by using Lemmata 4.1 to 4.3.

**Proposition A.3.** *Let $f_1$ and $f_2$ be jointly orientation-equivariant mappings according to Definition 4.1. Then $f_1 \circ_{equ} f_2 = f_1(f_2(\boldsymbol{X}_{equ}, \boldsymbol{X}_{inv}, \mathcal{G}, \mathfrak{O}), \boldsymbol{X}_{inv}, \mathcal{G}, \mathfrak{O})$ is also jointly orientation-equivariant.*

*Proof.*

$$
\begin{aligned}
(f_1 \circ_{\text{equ}} f_2)(\Delta_{\mathfrak{O}, \hat{\mathfrak{O}}} \boldsymbol{X}_{\text{equ}}, \boldsymbol{X}_{\text{inv}}, \mathcal{G}, \hat{\mathfrak{O}}) &= f_1(f_2(\Delta_{\mathfrak{O}, \hat{\mathfrak{O}}} \boldsymbol{X}_{\text{equ}}, \boldsymbol{X}_{\text{inv}}, \mathcal{G}, \hat{\mathfrak{O}}), \boldsymbol{X}_{\text{inv}}, \mathcal{G}, \hat{\mathfrak{O}}) \\
&= f_1(\Delta_{\mathfrak{O}, \hat{\mathfrak{O}}} f_2(\boldsymbol{X}_{\text{equ}}, \boldsymbol{X}_{\text{inv}}, \mathcal{G}, \mathfrak{O}), \boldsymbol{X}_{\text{inv}}, \mathcal{G}, \hat{\mathfrak{O}}) \\
&= \Delta_{\mathfrak{O}, \hat{\mathfrak{O}}} f_1(f_2(\boldsymbol{X}_{\text{equ}}, \boldsymbol{X}_{\text{inv}}, \mathcal{G}, \mathfrak{O}), \boldsymbol{X}_{\text{inv}}, \mathcal{G}, \mathfrak{O}) \\
&= \Delta_{\mathfrak{O}, \hat{\mathfrak{O}}} (f_1 \circ_{\text{equ}} f_2)(\boldsymbol{X}_{\text{equ}}, \mathcal{G}, \mathfrak{O})
\end{aligned}
$$

$\qquad \square$

Next we discuss the relationship between sign-equivariant and jointly orientation-equivariant mappings.

**Proposition A.4.** *Let $\sigma$ be a sign-equivariant mapping, i.e. $\sigma(-x, -y) = -\sigma(x, y)$ and $f_1$ and $f_2$ be jointly orientation-equivariant mappings. Then $f(\boldsymbol{X}_{equ}, \boldsymbol{X}_{inv}, \mathcal{G}, \mathfrak{O}) = \sigma(f_1(\boldsymbol{X}_{equ}, \boldsymbol{X}_{inv}, \mathcal{G}, \mathfrak{O}), f_2(\boldsymbol{X}_{equ}, \boldsymbol{X}_{inv}, \mathcal{G}, \mathfrak{O}))$ is a jointly orientation-equivariant mapping as well.*

*Proof.*

$$
\begin{aligned}
f(\Delta_{\mathfrak{O}, \hat{\mathfrak{O}}} \boldsymbol{X}_{\text{equ}}, \mathcal{G}, \hat{\mathfrak{O}}) &= \sigma(f_1(\Delta_{\mathfrak{O}, \hat{\mathfrak{O}}} \boldsymbol{X}_{\text{equ}}, \boldsymbol{X}_{\text{inv}}, \mathcal{G}, \hat{\mathfrak{O}}), f_2(\Delta_{\mathfrak{O}, \hat{\mathfrak{O}}} \boldsymbol{X}_{\text{equ}}, \boldsymbol{X}_{\text{inv}}, \mathcal{G}, \hat{\mathfrak{O}})) \\
&= \sigma(\Delta_{\mathfrak{O}, \hat{\mathfrak{O}}} f_1(\boldsymbol{X}_{\text{equ}}, \boldsymbol{X}_{\text{inv}}, \mathcal{G}, \mathfrak{O}), \Delta_{\mathfrak{O}, \hat{\mathfrak{O}}} f_2(\boldsymbol{X}_{\text{equ}}, \boldsymbol{X}_{\text{inv}}, \mathcal{G}, \mathfrak{O})) \\
&= \Delta_{\mathfrak{O}, \hat{\mathfrak{O}}} \sigma(f_1(\boldsymbol{X}_{\text{equ}}, \boldsymbol{X}_{\text{inv}}, \mathcal{G}, \mathfrak{O}), f_2(\boldsymbol{X}_{\text{equ}}, \boldsymbol{X}_{\text{inv}}, \mathcal{G}, \mathfrak{O})) \\
&= \Delta_{\mathfrak{O}, \hat{\mathfrak{O}}} f(\boldsymbol{X}_{\text{equ}}, \boldsymbol{X}_{\text{inv}}, \mathcal{G}, \mathfrak{O})
\end{aligned}
$$

$\qquad \square$

Note that Proposition A.4 generalizes to sign equivariant functions with one argument by defining $\sigma'(x) = \sigma(x, 0)$.

We can now prove that $\boldsymbol{H}_{\text{equ}}^{(l)}$ as defined in Equation (2) is a jointly equivariant mapping.

**Lemma A.5.** $\boldsymbol{H}_{equ}^{(l)}$ *as defined in Equation (2) is a jointly orientation-equivariant mapping.* $\boldsymbol{H}_{inv}^{(l)}$ *as defined in Equation (3) is a jointly orientation-invariant mapping.*

*Proof.* The proof is done inductively. Therefore, assume $\boldsymbol{H}_{\text{equ}}^{(l-1)}$ to be a jointly orientation-equivariant mapping and $\boldsymbol{H}_{\text{inv}}^{(l-1)}$ to be a jointly orientation-invariant mapping. For notational simplicity, we absorb the dependency on $\boldsymbol{X}_{\text{equ}}$, $\boldsymbol{X}_{\text{inv}}$, $\mathcal{G}$ and $\mathfrak{O}$ into the mapping itself: That is, for a mapping $\boldsymbol{F}$, we denote $\boldsymbol{F}(\boldsymbol{X}_{\text{equ}}, \boldsymbol{X}_{\text{inv}}, \mathcal{G}, \mathfrak{O}) = \boldsymbol{F}$ and $\boldsymbol{F}(\Delta_{\mathfrak{O},\hat{\mathfrak{O}}} \boldsymbol{X}_{\text{equ}}, \boldsymbol{X}_{\text{inv}}, \mathcal{G}, \hat{\mathfrak{O}}) = \hat{\boldsymbol{F}}$.

We first show that $\boldsymbol{Z}_{\text{equ}}^{(l)}$ to be a jointly orientation-equivariant mapping.

$$\begin{aligned}
\hat{\boldsymbol{Z}}_{\text{equ}}^{(l)} &= \sigma_{\text{equ}}(f_{\text{equ}}^{(l)}(\hat{\boldsymbol{H}}_{\text{equ}}^{(l-1)})\boldsymbol{W}_{\text{equ}\to\text{equ}}^{(l)} + f_{\text{inv}\to\text{equ}}^{(l)}(\hat{\boldsymbol{H}}_{\text{inv}}^{(l-1)})\boldsymbol{W}_{\text{inv}\to\text{equ}}^{(l)} + \hat{\boldsymbol{H}}_{\text{equ}}^{(l-1)}\boldsymbol{W}_{\text{equ}}^{(l)}) \\
&= \sigma_{\text{equ}}(f_{\text{equ}}^{(l)}(\Delta_{\mathfrak{O},\hat{\mathfrak{O}}}\boldsymbol{H}_{\text{equ}}^{(l-1)})\boldsymbol{W}_{\text{equ}\to\text{equ}}^{(l)} + f_{\text{inv}\to\text{equ}}^{(l)}(\boldsymbol{H}_{\text{inv}}^{(l-1)})\boldsymbol{W}_{\text{inv}\to\text{equ}}^{(l)} + \Delta_{\mathfrak{O},\hat{\mathfrak{O}}}\boldsymbol{H}_{\text{equ}}^{(l-1)}\boldsymbol{W}_{\text{equ}}^{(l)}) \\
&= \sigma_{\text{equ}}(\Delta_{\mathfrak{O},\hat{\mathfrak{O}}}f_{\text{equ}}^{(l)}(\boldsymbol{H}_{\text{equ}}^{(l-1)})\boldsymbol{W}_{\text{equ}\to\text{equ}}^{(l)} + \Delta_{\mathfrak{O},\hat{\mathfrak{O}}}f_{\text{inv}\to\text{equ}}^{(l)}(\boldsymbol{H}_{\text{inv}}^{(l-1)})\boldsymbol{W}_{\text{inv}\to\text{equ}}^{(l)} + \Delta_{\mathfrak{O},\hat{\mathfrak{O}}}\boldsymbol{H}_{\text{equ}}^{(l-1)}\boldsymbol{W}_{\text{equ}}^{(l)}) \\
&= \Delta_{\mathfrak{O},\hat{\mathfrak{O}}}\sigma_{\text{equ}}(f_{\text{equ}}^{(l)}(\boldsymbol{H}_{\text{equ}}^{(l-1)})\boldsymbol{W}_{\text{equ}\to\text{equ}}^{(l)} + f_{\text{inv}\to\text{equ}}^{(l)}(\boldsymbol{H}_{\text{inv}}^{(l-1)})\boldsymbol{W}_{\text{inv}\to\text{equ}}^{(l)} + \boldsymbol{H}_{\text{equ}}^{(l-1)}\boldsymbol{W}_{\text{equ}}^{(l)}) \\
&= \Delta_{\mathfrak{O},\hat{\mathfrak{O}}}\boldsymbol{Z}_{\text{equ}}^{(l)}
\end{aligned}$$

Here, we first used the induction assumption, then applied the joint orientation equivariance property of $\boldsymbol{L}_{\text{equ}}^{(q)}$ and $\boldsymbol{L}_{\text{inv}\to\text{equ}}^{(q)}$ proven in Lemmata 4.1 and 4.3 and lastly used the fact that we assume $\sigma_{\text{equ}}$ to be sign-invariant function to apply Proposition A.4.

Similarily, we can show that $\boldsymbol{Z}_{\text{equ}}^{(l)}$ to be a jointly orientation-equivariant mapping.

$$\begin{aligned}
\hat{\boldsymbol{Z}}_{\text{inv}}^{(l)} &= \sigma_{\text{inv}}(f_{\text{inv}}^{(l)}(\hat{\boldsymbol{H}}_{\text{inv}}^{(l-1)})\boldsymbol{W}_{\text{inv}\to\text{inv}}^{(l)} + f_{\text{equ}\to\text{inv}}^{(l)}(\hat{\boldsymbol{H}}_{\text{equ}}^{(l-1)})\boldsymbol{W}_{\text{equ}\to\text{inv}}^{(l)} + \hat{\boldsymbol{H}}_{\text{inv}}^{(l-1)}\boldsymbol{W}_{\text{inv}}^{(l)}) \\
&= \sigma_{\text{inv}}(f_{\text{inv}}^{(l)}(\boldsymbol{H}_{\text{inv}}^{(l-1)})\boldsymbol{W}_{\text{inv}\to\text{inv}}^{(l)} + f_{\text{equ}\to\text{inv}}^{(l)}(\Delta_{\mathfrak{O},\hat{\mathfrak{O}}}\boldsymbol{H}_{\text{equ}}^{(l-1)})\boldsymbol{W}_{\text{equ}\to\text{inv}}^{(l)} + \boldsymbol{H}_{\text{inv}}^{(l-1)}\boldsymbol{W}_{\text{inv}}^{(l)}) \\
&= \sigma_{\text{inv}}(f_{\text{inv}}^{(l)}(\boldsymbol{H}_{\text{inv}}^{(l-1)})\boldsymbol{W}_{\text{inv}\to\text{inv}}^{(l)} + f_{\text{equ}\to\text{inv}}^{(l)}(\boldsymbol{H}_{\text{equ}}^{(l-1)})\boldsymbol{W}_{\text{equ}\to\text{inv}}^{(l)} + \boldsymbol{H}_{\text{inv}}^{(l-1)}\boldsymbol{W}_{\text{inv}}^{(l)}) \\
&= \boldsymbol{Z}_{\text{inv}}^{(l)}
\end{aligned}$$

Here, we first used the induction assumption, then applied the joint orientation invariance property of $\boldsymbol{L}_{\text{inv}}^{(q)}$ and $\boldsymbol{L}_{\text{equ}\to\text{inv}}^{(q)}$ proven in Lemmata 4.2 and 4.3.

Next, we can show that $\boldsymbol{H}_{\text{equ}}^{(l)}$ is an orientation-equivariant mapping.

$$\begin{aligned}
\hat{\boldsymbol{H}}_{\text{equ}}^{(l)} &= \sigma_{\text{equ}}(\hat{\boldsymbol{Z}}_{\text{equ}}^{(l)}\boldsymbol{W}_{\text{F,equ,equ}}^{(l)} \odot \hat{\boldsymbol{Z}}_{\text{inv}}^{(l)}\boldsymbol{W}_{\text{F,equ,inv}}^{(l)} + \hat{\boldsymbol{Z}}_{\text{equ}}^{(l)}) \\
&= \sigma_{\text{equ}}((\Delta_{\mathfrak{O},\hat{\mathfrak{O}}}\boldsymbol{Z}_{\text{equ}}^{(l)}\boldsymbol{W}_{\text{F,equ,equ}}^{(l)}) \odot \boldsymbol{Z}_{\text{inv}}^{(l)}\boldsymbol{W}_{\text{F,equ,inv}}^{(l)} + \Delta_{\mathfrak{O},\hat{\mathfrak{O}}}\boldsymbol{Z}_{\text{equ}}^{(l)}) \\
&= \sigma_{\text{equ}}(\Delta_{\mathfrak{O},\hat{\mathfrak{O}}}(\boldsymbol{Z}_{\text{equ}}^{(l)}\boldsymbol{W}_{\text{F,equ,equ}}^{(l)} \odot \boldsymbol{Z}_{\text{inv}}^{(l)}\boldsymbol{W}_{\text{F,equ,inv}}^{(l)}) + \Delta_{\mathfrak{O},\hat{\mathfrak{O}}}\boldsymbol{Z}_{\text{equ}}^{(l)}) \\
&= \Delta_{\mathfrak{O},\hat{\mathfrak{O}}}\sigma_{\text{equ}}((\boldsymbol{Z}_{\text{equ}}^{(l)}\boldsymbol{W}_{\text{F,equ,equ}}^{(l)} \odot \boldsymbol{Z}_{\text{inv}}^{(l)}\boldsymbol{W}_{\text{F,equ,inv}}^{(l)}) + \boldsymbol{Z}_{\text{equ}}^{(l)}) \\
&= \Delta_{\mathfrak{O},\hat{\mathfrak{O}}}\boldsymbol{H}_{\text{equ}}^{(l)}
\end{aligned}$$

We first use the previously proven joint orientation equivariance of $\boldsymbol{Z}_{\text{equ}}^{(l)}$, and then the fact both $\Delta_{\mathfrak{O},\hat{\mathfrak{O}}}$ and $\odot$ are element-wise multiplications and, therefore, are associative operations. Lastly, we again make use of Proposition A.4 for the sign-equivariant function $\sigma_{\text{equ}}$.

Lastly, we can show that $\boldsymbol{H}_{\text{inv}}^{(l)}$ is an orientation-invariant mapping.

$$
\begin{aligned}
\hat{\boldsymbol{H}}_{\text{inv}}^{(l)} &= \sigma_{\text{inv}}(\hat{\boldsymbol{Z}}_{\text{inv}}^{(l)} \boldsymbol{W}_{\text{F,inv,inv}}^{(l)} \odot \text{abs}(\hat{\boldsymbol{Z}}_{\text{equ}}^{(l)} \boldsymbol{W}_{\text{F,inv,equ}}^{(l)}) + \hat{\boldsymbol{Z}}_{\text{inv}}^{(l)}) \\
&= \sigma_{\text{inv}}(\boldsymbol{Z}_{\text{inv}}^{(l)} \boldsymbol{W}_{\text{F,inv,inv}}^{(l)} \odot \text{abs}(\Delta_{\mathfrak{O},\hat{\mathfrak{O}}} \boldsymbol{Z}_{\text{equ}}^{(l)} \boldsymbol{W}_{\text{F,inv,equ}}^{(l)}) + \boldsymbol{Z}_{\text{inv}}^{(l)}) \\
&= \sigma_{\text{inv}}(\boldsymbol{Z}_{\text{inv}}^{(l)} \boldsymbol{W}_{\text{F,inv,inv}}^{(l)} \odot \text{abs}(\boldsymbol{Z}_{\text{equ}}^{(l)} \boldsymbol{W}_{\text{F,inv,equ}}^{(l)}) + \boldsymbol{Z}_{\text{inv}}^{(l)}) \\
&= \boldsymbol{H}_{\text{inv}}^{(l)}
\end{aligned}
$$

Here, we first use the previously proven joint orientation invariance of $\boldsymbol{Z}_{\text{inv}}^{(l)}$. Then, we notice that $\Delta_{\mathfrak{O},\hat{\mathfrak{O}}}$ corresponds to an element-wise multiplication with $\pm 1$ and is therefore canceled out by the element-wise absolute value function. $\qquad\square$

## A.2 PERMUTATION EQUIVARIANCE

A common requirement for GNNs is equivariance with respect to the ordering of input to enable generalization (Maron et al., 2018; Wu et al., 2021). While node-level GNNs are equivariant with respect to node permutations, we require edge-level permutation equivariance.

**Definition A.1** (Permutation Equivariance). Let $\mathfrak{O}$ be an orientation of edges on $\mathcal{G}$ and $\boldsymbol{P} \in \mathbb{R}^{m \times m}$ be a permutation matrix on the edges. Let $\mathfrak{O}_{\boldsymbol{P}}$ and $\mathcal{G}_{\boldsymbol{P}}$ be its application to $\mathfrak{O}$ and $\mathcal{G}$ respectively. We say that a mapping $f$ satisfies permutation equivariance if for any orientation-equivariant signal $\boldsymbol{X}_{\text{equ}} \in \mathbb{C}^{m \times d_{\text{equ}}}$ and any orientation-invariant signal $\boldsymbol{X}_{\text{inv}} \in \mathbb{C}^{m \times d_{\text{inv}}}$:

$$
\boldsymbol{P} f(\boldsymbol{X}_{\text{equ}}, \boldsymbol{X}_{\text{inv}}, \mathcal{G}, \mathfrak{O}) = f(\boldsymbol{P}\boldsymbol{X}_{\text{equ}}, \boldsymbol{P}\boldsymbol{X}_{\text{inv}}, \mathcal{G}_{\boldsymbol{P}}, \mathfrak{O}_{\boldsymbol{P}}).
$$

We now proceed to prove that EIGN is a permutation equivariant mapping according to Definition A.1. Again, we first concern boundary operators and show results for the generalized, directed case and recover Equations (4) and (5) with $q = 0$.

**Proposition A.6.** $\boldsymbol{L}_{equ}^{(q)}$ induces a permutation equivariant mapping according to Definition A.1.

*Proof.* Let $\mathfrak{O}$ be an orientation of edges on $\mathcal{G}$ and $\boldsymbol{P}$ be a permutation matrix. Let $\mathfrak{O}_{\boldsymbol{P}}$ and $\mathcal{G}_{\boldsymbol{P}}$ its application to $\mathfrak{O}$ and $\mathcal{G}$ respectively.

$$
\begin{aligned}
f(\boldsymbol{P}\boldsymbol{X}_{\text{equ}}, \boldsymbol{P}\boldsymbol{X}_{\text{inv}}, \mathcal{G}_{\boldsymbol{P}}, \mathfrak{O}_{\boldsymbol{P}}) &= (\boldsymbol{B}_{\text{equ}}^{(q)}(\mathcal{G}_{\boldsymbol{P}}, \mathfrak{O}_{\boldsymbol{P}}))^H h(\boldsymbol{B}_{\text{equ}}^{(q)}(\mathcal{G}_{\boldsymbol{P}}, \mathfrak{O}_{\boldsymbol{P}}) \boldsymbol{P}\boldsymbol{X}_{\text{equ}}) \\
&= (\boldsymbol{B}_{\text{equ}}^{(q)}(\mathcal{G}, \mathfrak{O}) \boldsymbol{P}^H)^H h(\boldsymbol{B}_{\text{equ}}^{(q)}(\mathcal{G}, \mathfrak{O}) \boldsymbol{P}^H \boldsymbol{P}\boldsymbol{X}_{\text{equ}}) \\
&= \boldsymbol{P}(\boldsymbol{B}_{\text{equ}}^{(q)}(\mathcal{G}, \mathfrak{O}))^H h(\boldsymbol{B}_{\text{equ}}^{(q)}(\mathcal{G}, \mathfrak{O}) \boldsymbol{X}_{\text{equ}}) \\
&= \boldsymbol{P} f(\boldsymbol{X}_{\text{equ}}, \boldsymbol{X}_{\text{inv}}, \mathcal{G}, \mathfrak{O})
\end{aligned}
$$

$\qquad\square$

**Proposition A.7.** $\boldsymbol{L}_{inv}^{(q)}$ induces a permutation equivariant mapping according to Definition A.1.

*Proof.* Let $\mathfrak{O}$ be an orientation of edges on $\mathcal{G}$ and $\boldsymbol{P}$ be a permutation matrix. Let $\mathfrak{O}_{\boldsymbol{P}}$ and $\mathcal{G}_{\boldsymbol{P}}$ its application to $\mathfrak{O}$ and $\mathcal{G}$ respectively.

$$
\begin{aligned}
g(\boldsymbol{P}\boldsymbol{X}_{\text{inv}}, \boldsymbol{P}\boldsymbol{X}_{\text{inv}}, \mathcal{G}_{\boldsymbol{P}}, \mathfrak{O}_{\boldsymbol{P}}) &= (\boldsymbol{B}_{\text{inv}}^{(q)}(\mathcal{G}_{\boldsymbol{P}}, \mathfrak{O}_{\boldsymbol{P}}))^H h(\boldsymbol{B}_{\text{inv}}^{(q)}(\mathcal{G}_{\boldsymbol{P}}, \mathfrak{O}_{\boldsymbol{P}}) \boldsymbol{P}\boldsymbol{X}_{\text{inv}}) \\
&= (\boldsymbol{B}_{\text{inv}}^{(q)}(\mathcal{G}, \mathfrak{O}) \boldsymbol{P}^H)^H h(\boldsymbol{B}_{\text{inv}}^{(q)}(\mathcal{G}, \mathfrak{O}) \boldsymbol{P}^H \boldsymbol{P}\boldsymbol{X}_{\text{inv}}) \\
&= \boldsymbol{P}(\boldsymbol{B}_{\text{inv}}^{(q)}(\mathcal{G}, \mathfrak{O}))^H h(\boldsymbol{B}_{\text{inv}}^{(q)}(\mathcal{G}, \mathfrak{O}) \boldsymbol{X}_{\text{inv}}) \\
&= \boldsymbol{P} g(\boldsymbol{X}_{\text{inv}}, \boldsymbol{X}_{\text{inv}}, \mathcal{G}, \mathfrak{O})
\end{aligned}
$$

$\qquad\square$

**Proposition A.8.** $\boldsymbol{L}_{equ \to inv}^{(q)}$ induces a permutation equivariant mapping according to Definition A.1.

*Proof.* Let $\mathfrak{O}$ be an orientation of edges on $\mathcal{G}$ and $P$ be a permutation matrix. Let $\mathfrak{O}_P$ and $\mathcal{G}_P$ its application to $\mathfrak{O}$ and $\mathcal{G}$ respectively.

$$
\begin{aligned}
f(P X_{\text{equ}}, P X_{\text{inv}}, \mathcal{G}_P, \mathfrak{O}_P) &= (B_{\text{inv}}^{(q)}(\mathcal{G}_P, \mathfrak{O}_P))^H h(B_{\text{equ}}^{(q)}(\mathcal{G}_P, \mathfrak{O}_P) P X_{\text{equ}}) \\
&= (B_{\text{inv}}^{(q)}(\mathcal{G}, \mathfrak{O}) P^H)^H h(B_{\text{equ}}^{(q)}(\mathcal{G}, \mathfrak{O}) P^H P X_{\text{equ}}) \\
&= P(B_{\text{inv}}^{(q)}(\mathcal{G}, \mathfrak{O}))^H h(B_{\text{equ}}^{(q)}(\mathcal{G}, \mathfrak{O}) X_{\text{equ}}) \\
&= P f(X_{\text{equ}}, X_{\text{inv}}, \mathcal{G}, \mathfrak{O})
\end{aligned}
$$

$\square$

**Proposition A.9.** $L_{inv \to equ}^{(q)}$ *induces a permutation equivariant mapping according to Definition A.1.*

*Proof.* Let $\mathfrak{O}$ be an orientation of edges on $\mathcal{G}$ and $P$ be a permutation matrix. Let $\mathfrak{O}_P$ and $\mathcal{G}_P$ its application to $\mathfrak{O}$ and $\mathcal{G}$ respectively.

$$
\begin{aligned}
f(P X_{\text{equ}}, P X_{\text{inv}}, \mathcal{G}_P, \mathfrak{O}_P) &= (B_{\text{equ}}^{(q)}(\mathcal{G}_P, \mathfrak{O}_P))^H h(B_{\text{inv}}^{(q)}(\mathcal{G}_P, \mathfrak{O}_P) P X_{\text{inv}}) \\
&= (B_{\text{equ}}^{(q)}(\mathcal{G}, \mathfrak{O}) P^H)^H h(B_{\text{inv}}^{(q)}(\mathcal{G}, \mathfrak{O}) P^H P X_{\text{inv}}) \\
&= P(B_{\text{equ}}^{(q)}(\mathcal{G}, \mathfrak{O}))^H h(B_{\text{inv}}^{(q)}(\mathcal{G}, \mathfrak{O}) X_{\text{inv}}) \\
&= P f(X_{\text{equ}}, X_{\text{inv}}, \mathcal{G}, \mathfrak{O})
\end{aligned}
$$

$\square$

**Lemma A.10.** $H_{equ}^{(l)}$ *and* $H_{inv}^{(l)}$ *as defined in Equation* (2) *are permutation equivariant mappings.*

*Proof.* The proof is done inductively. Therefore, assume $H_{\text{equ}}^{(l-1)}$ and $H_{\text{inv}}^{(l-1)}$ to be a permutation equivariant mappings. For simplicity, we, again, absorb the dependency on $X_{\text{equ}}$, $X_{\text{inv}}$, $\mathcal{G}$ and $\mathfrak{O}$ into the mapping itself: That is, for a mapping $F$, we denote $F(X_{\text{equ}}, X_{\text{inv}}, \mathcal{G}, \mathfrak{O}) = F$ and $F(P X_{\text{equ}}, P X_{\text{inv}}, \mathcal{G}_P, \mathfrak{O}_P) = \tilde{F}$.

We first show that $Z_{\text{equ}}^{(l)}$ to be a permutation equivariant mapping.

$$
\begin{aligned}
\tilde{Z}_{\text{equ}}^{(l)} &= \sigma_{\text{equ}}(f_{\text{equ}}^{(l)}(\tilde{H}_{\text{equ}}^{(l-1)}) W_{\text{equ}\to\text{equ}}^{(l)} + f_{\text{inv}\to\text{equ}}^{(l)}(\tilde{H}_{\text{inv}}^{(l-1)}) W_{\text{inv}\to\text{equ}}^{(l)} + \tilde{H}_{\text{equ}}^{(l-1)} W_{\text{equ}}^{(l)}) \\
&= \sigma_{\text{equ}}(f_{\text{equ}}^{(l)}(P H_{\text{equ}}^{(l-1)}) W_{\text{equ}\to\text{equ}}^{(l)} + f_{\text{inv}\to\text{equ}}^{(l)}(P H_{\text{inv}}^{(l-1)}) W_{\text{inv}\to\text{equ}}^{(l)} + P H_{\text{equ}}^{(l-1)} W_{\text{equ}}^{(l)}) \\
&= \sigma_{\text{equ}}(P f_{\text{equ}}^{(l)}(H_{\text{equ}}^{(l-1)}) W_{\text{equ}\to\text{equ}}^{(l)} + P f_{\text{inv}\to\text{equ}}^{(l)}(H_{\text{inv}}^{(l-1)}) W_{\text{inv}\to\text{equ}}^{(l)} + P H_{\text{equ}}^{(l-1)} W_{\text{equ}}^{(l)}) \\
&= P \sigma_{\text{equ}}(f_{\text{equ}}^{(l)}(H_{\text{equ}}^{(l-1)}) W_{\text{equ}\to\text{equ}}^{(l)} + f_{\text{inv}\to\text{equ}}^{(l)}(H_{\text{inv}}^{(l-1)}) W_{\text{inv}\to\text{equ}}^{(l)} + H_{\text{equ}}^{(l-1)} W_{\text{equ}}^{(l)}) \\
&= P Z_{\text{equ}}^{(l)}
\end{aligned}
$$

Here, we first used the induction assumption, then applied the permutation equivariance property of $L_{\text{equ}}^{(q)}$ and $L_{\text{inv}\to\text{equ}}^{(q)}$ proven in Propositions A.6 and A.9 and lastly used the fact that we assume $\sigma_{\text{equ}}$ to be an element-wise function which, hence, commutes with permutations.

Similarily, we can show that $Z_{\text{equ}}^{(l)}$ to be a permutation equivariant mapping.

$$
\begin{aligned}
\tilde{Z}_{\text{inv}}^{(l)} &= \sigma_{\text{inv}}(f_{\text{inv}}^{(l)}(\tilde{H}_{\text{inv}}^{(l-1)}) W_{\text{inv}\to\text{inv}}^{(l)} + f_{\text{equ}\to\text{inv}}^{(l)}(\tilde{H}_{\text{equ}}^{(l-1)}) W_{\text{equ}\to\text{inv}}^{(l)} + \tilde{H}_{\text{inv}}^{(l-1)} W_{\text{inv}}^{(l)}) \\
&= \sigma_{\text{inv}}(f_{\text{inv}}^{(l)}(P H_{\text{inv}}^{(l-1)}) W_{\text{inv}\to\text{inv}}^{(l)} + f_{\text{equ}\to\text{inv}}^{(l)}(P H_{\text{equ}}^{(l-1)}) W_{\text{equ}\to\text{inv}}^{(l)} + P H_{\text{inv}}^{(l-1)} W_{\text{inv}}^{(l)}) \\
&= \sigma_{\text{inv}}(P f_{\text{inv}}^{(l)}(H_{\text{inv}}^{(l-1)}) W_{\text{inv}\to\text{inv}}^{(l)} + P f_{\text{equ}\to\text{inv}}^{(l)}(H_{\text{equ}}^{(l-1)}) W_{\text{equ}\to\text{inv}}^{(l)} + P H_{\text{inv}}^{(l-1)} W_{\text{inv}}^{(l)}) \\
&= P \sigma_{\text{inv}}(f_{\text{inv}}^{(l)}(H_{\text{inv}}^{(l-1)}) W_{\text{inv}\to\text{inv}}^{(l)} + f_{\text{equ}\to\text{inv}}^{(l)}(H_{\text{equ}}^{(l-1)}) W_{\text{equ}\to\text{inv}}^{(l)} + H_{\text{inv}}^{(l-1)} W_{\text{inv}}^{(l)}) \\
&= P Z_{\text{inv}}^{(l)}
\end{aligned}
$$

Here, we first used the induction assumption, then applied the permutation equivariance property of $L_{\text{equ}}^{(q)}$ and $L_{\text{inv}\to\text{equ}}^{(q)}$ proven in Propositions A.7 and A.8 and lastly used the fact that we assume $\sigma_{\text{equ}}$ to be an element-wise function which, hence, commutes with permutations.

Next, we can show that $\boldsymbol{H}_{\text{equ}}^{(l)}$ is an permutation equivariant mapping.

$$
\begin{aligned}
\tilde{\boldsymbol{H}}_{\text{equ}}^{(l)} &= \sigma_{\text{equ}}(\tilde{\boldsymbol{Z}}_{\text{equ}}^{(l)}\boldsymbol{W}_{\text{F,equ,equ}}^{(l)} \odot \tilde{\boldsymbol{Z}}_{\text{inv}}^{(l)}\boldsymbol{W}_{\text{F,equ,inv}}^{(l)} + \tilde{\boldsymbol{Z}}_{\text{equ}}^{(l)}) \\
&= \sigma_{\text{equ}}((\boldsymbol{P}\boldsymbol{Z}_{\text{equ}}^{(l)}\boldsymbol{W}_{\text{F,equ,equ}}^{(l)}) \odot (\boldsymbol{P}\boldsymbol{Z}_{\text{inv}}^{(l)}\boldsymbol{W}_{\text{F,equ,inv}}^{(l)}) + \boldsymbol{P}\boldsymbol{Z}_{\text{equ}}^{(l)}) \\
&= \sigma_{\text{equ}}(\boldsymbol{P}(\boldsymbol{Z}_{\text{equ}}^{(l)}\boldsymbol{W}_{\text{F,equ,equ}}^{(l)} \odot \boldsymbol{Z}_{\text{inv}}^{(l)}\boldsymbol{W}_{\text{F,equ,inv}}^{(l)}) + \boldsymbol{P}\boldsymbol{Z}_{\text{equ}}^{(l)}) \\
&= \boldsymbol{P}\sigma_{\text{equ}}(\boldsymbol{Z}_{\text{equ}}^{(l)}\boldsymbol{W}_{\text{F,equ,equ}}^{(l)} \odot \boldsymbol{Z}_{\text{inv}}^{(l)}\boldsymbol{W}_{\text{F,equ,inv}}^{(l)} + \boldsymbol{Z}_{\text{equ}}^{(l)}) \\
&= \boldsymbol{P}\boldsymbol{H}_{\text{equ}}^{(l)}
\end{aligned}
$$

We first use the previously proven joint permutation equivariance of $\boldsymbol{Z}_{\text{equ}}^{(l)}$, and then the fact both $\sigma_{\text{equ}}$ and $\odot$ are element-wise operations and, therefore, commute with permutations.

Lastly, we can show that $\boldsymbol{H}_{\text{inv}}^{(l)}$ is an orientation-invariant mapping.

$$
\begin{aligned}
\tilde{\boldsymbol{H}}_{\text{inv}}^{(l)} &= \sigma_{\text{inv}}(\tilde{\boldsymbol{Z}}_{\text{inv}}^{(l)}\boldsymbol{W}_{\text{F,inv,inv}}^{(l)} \odot \text{abs}(\tilde{\boldsymbol{Z}}_{\text{equ}}^{(l)}\boldsymbol{W}_{\text{F,inv,equ}}^{(l)}) + \tilde{\boldsymbol{Z}}_{\text{inv}}^{(l)}) \\
&= \sigma_{\text{inv}}((\boldsymbol{P}\boldsymbol{Z}_{\text{inv}}^{(l)}\boldsymbol{W}_{\text{F,inv,inv}}^{(l)}) \odot (\text{abs}(\boldsymbol{P}\boldsymbol{Z}_{\text{equ}}^{(l)}\boldsymbol{W}_{\text{F,inv,equ}}^{(l)})) + \boldsymbol{P}\boldsymbol{Z}_{\text{inv}}^{(l)}) \\
&= \sigma_{\text{inv}}((\boldsymbol{P}\boldsymbol{Z}_{\text{inv}}^{(l)}\boldsymbol{W}_{\text{F,inv,inv}}^{(l)}) \odot (\boldsymbol{P}\,\text{abs}(\boldsymbol{Z}_{\text{equ}}^{(l)}\boldsymbol{W}_{\text{F,inv,equ}}^{(l)})) + \boldsymbol{P}\boldsymbol{Z}_{\text{inv}}^{(l)}) \\
&= \sigma_{\text{inv}}(\boldsymbol{P}(\boldsymbol{Z}_{\text{inv}}^{(l)}\boldsymbol{W}_{\text{F,inv,inv}}^{(l)} \odot \text{abs}(\boldsymbol{Z}_{\text{equ}}^{(l)}\boldsymbol{W}_{\text{F,inv,equ}}^{(l)})) + \boldsymbol{P}\boldsymbol{Z}_{\text{inv}}^{(l)}) \\
&= \boldsymbol{P}\sigma_{\text{inv}}(\boldsymbol{Z}_{\text{inv}}^{(l)}\boldsymbol{W}_{\text{F,inv,inv}}^{(l)} \odot \text{abs}(\boldsymbol{Z}_{\text{equ}}^{(l)}\boldsymbol{W}_{\text{F,inv,equ}}^{(l)}) + \boldsymbol{Z}_{\text{inv}}^{(l)}) \\
&= \boldsymbol{P}\boldsymbol{H}_{\text{inv}}^{(l)}
\end{aligned}
$$

Here, we first use the previously proven joint permutation equivariance of $\boldsymbol{Z}_{\text{inv}}^{(l)}$, and then the fact $\sigma_{\text{inv}}$, abs and $\odot$ are element-wise operations and, therefore, commute with permutations. $\square$

## A.3 THE MAIN RESULT

We can now plug Lemmas A.5 and A.10 together and prove Theorem 4.1.

**Theorem 4.1.** *(i) $\boldsymbol{H}_{equ}^{(L)}$ is a jointly orientation-equivariant mapping. (ii) $\boldsymbol{H}_{inv}^{(L)}$ is a jointly orientation-invariant mapping. (iii) Both $\boldsymbol{H}_{equ}^{(L)}$ and $\boldsymbol{H}_{inv}^{(L)}$ are permutation equivariant mappings.*

*Proof.* The proof of (i) and (ii) is directly given by Lemma A.5. The proof of (iii) is directly given by Lemma A.10. $\square$

## A.4 INTER-MODALITY CONVOLUTIONS TO LEARN ORIENTATION-EQUIVARIANT REPRESENTATIONS FROM ORIENTATION-INVARIANT REPRESENTATIONS

Here, we show that not using the inter-modality convolutions $f_{\text{inv}\to\text{equ}}^{(l)}$ results in the model outputting for the orientation-equivariant edge representations in case there are no orientation-equivariant inputs available ($\boldsymbol{X}_{\text{equ}} = \boldsymbol{0}$). Therefore, omitting these convolutions prevents the model from predicting any non-zero orientation-equivariant output for undirected edges if orientation-equivariant inputs are unavailable.

**Lemma A.11.** *Computing $\boldsymbol{Z}_{equ}^{(l)}$ as per Equation (2) but without using $f_{inv\to equ)}^{(l)}$ implies that if $\boldsymbol{H}_{equ}^{(l-1)} = \boldsymbol{0}$, then also $(\boldsymbol{Z}_{equ}^{(l)})_{\mathcal{E}_U} = \boldsymbol{0}$ for undirected edges.*

*Proof.*

$$
\begin{aligned}
(\boldsymbol{Z}_{\text{equ}}^{(l)})_{\mathcal{E}_{\text{U}}} &= \sigma_{\text{equ}}(f_{\text{equ}}^{(l)}(\boldsymbol{H}_{\text{equ}}^{(l-1)})\boldsymbol{W}_{\text{equ}\to\text{equ}}^{(l)} + \boldsymbol{H}_{\text{equ}}^{(l-1)}\boldsymbol{W}_{\text{equ}}^{(l)})_{\mathcal{E}_{\text{U}}} \\
&= \sigma_{\text{equ}}(f_{\text{equ}}^{(l)}(\boldsymbol{0})\boldsymbol{W}_{\text{equ}\to\text{equ}}^{(l)} + \boldsymbol{0}\boldsymbol{W}_{\text{equ}}^{(l)})_{\mathcal{E}_{\text{U}}} \\
&= \sigma_{\text{equ}}((\boldsymbol{B}_{\text{equ}}^{(q)})^{H} h(\boldsymbol{B}_{\text{equ}}^{(q)}\boldsymbol{0})\boldsymbol{W}_{\text{equ}\to\text{equ}}^{(l)})_{\mathcal{E}_{\text{U}}} \\
&= \sigma_{\text{equ}}((\boldsymbol{B}_{\text{equ}}^{(q)})^{H} h(\boldsymbol{0})\boldsymbol{W}_{\text{equ}\to\text{equ}}^{(l)})_{\mathcal{E}_{\text{U}}} \\
&= \sigma_{\text{equ}}((\boldsymbol{B}_{\text{equ}}^{(q)})_{\mathcal{E}_{\text{U}}}^{H} h(\boldsymbol{0})\boldsymbol{W}_{\text{equ}\to\text{equ}}^{(l)}) \\
&= \sigma_{\text{equ}}((\boldsymbol{B}_{\text{equ}}^{(q)})_{\mathcal{E}_{\text{U}}}^{H} \boldsymbol{c})\boldsymbol{W}_{\text{equ}\to\text{equ}}^{(l)}) \\
&= \sigma_{\text{equ}}(\boldsymbol{0}\boldsymbol{W}_{\text{equ}\to\text{equ}}^{(l)}) \\
&= \sigma_{\text{equ}}(\boldsymbol{0})) \\
&= \boldsymbol{0}
\end{aligned}
$$

$\square$

Here, we have used that the node feature mapping $h$ will get zeros as input and therefore produce the same constant embedding $\boldsymbol{c}$ for all nodes. Along undirected edges, the flow that is induced by $(\boldsymbol{B}_{\text{equ}}^{(q)})^{H}$ corresponds to the differences between its endpoints, which consequently will be zero as well. Note that similarly each directed edge will be assigned the representation $(\exp(i\pi q) - \exp(-i\pi q))\boldsymbol{c}$. While being non-zero, this layer architecture is only able to produce a constant feature for both directed and undirected edges which heavily limits its expressivity. This is also well reflected in Table 15.

## B  CONSIDERATIONS FOR MODELLING ORIENTATION EQUIVARIANT AND ORIENTATION INVARIANT SIGNALS

Here, we provide additional considerations for how orientation equivariant and invariant signals should be modeled on graphs that contain both directed and undirected edges. This discussion supplements Figure 2. We begin by highlighting the limitations of previous architectures that treat all edge signals as fully orientation invariant or orientation equivariant respectively.

**Orientation Invariant Architectures.** Most architectures that are not grounded in Algebraic Topology treat all inputs and targets as orientation invariant. That is, the input features and the output features are assumed to not have an inherent direction. This is, however, limiting, as in many real-world applications (traffic, electrical engineering, water flow networks, etc.) accounting for the direction of a signal is crucial. Therefore, these architectures are not applicable to these settings.

**Orientation Equivariant Architectures.** Note that it is impossible to represent the direction of a signal through a scalar value without defining a reference orientation for the associated edge: Either the signal direction matches the reference orientation or it does not. For example, topological methods use the sign of a signal to indicate if its direction matches the reference orientation. In the case of directed edges, there is a clear choice for defining this reference. For undirected edges, however, an arbitrary reference must be fixed which is typically done in topological approaches. Since the reference orientation is used only to represent direction in signals and can be seen as a representational basis it should not affect the predictions of a model. This leads to an equivariance condition for these architectures: If the reference orientation changes, inputs, and outputs of a model should be represented with respect to this new orientation but not change apart from that. Topological models typically satisfy this notion (Roddenberry & Segarra, 2019; Roddenberry et al., 2021).

They, however, treat all inputs and outputs as signals that have inherent direction, i.e. represent them with respect to an orientation. Therefore, they can not model information that does not come with inherent direction like the number of lanes in a street, resistance, or pipe diameter. Enforcing orientation equivariance for all inputs and outputs of a model does not apply to settings where also orientation invariant signals that are irrespective of a reference orientation are available. In particular, these models require representing input and output signals with respect to an orientation and it is

ill-defined to define how an orientation invariant signal should be represented with respect to a reference orientation, e.g. through its sign.

**Edge Direction.** In addition to the inherent direction to the signal of an edge, the edge itself can be directed. In practice, the direction of an edge implies constraints on the solutions admissible to a problem: One-way streets in traffic networks prohibit a certain traffic flow direction, and diodes and valves play a similar role in electrical circuits or pipe networks. However, orientation equivariant models are insensitive to the (arbitrarily chosen) orientation of edge edge. Therefore, even if the orientation of directed edges is fixed to coincide with their direction, this information can not be utilized by a fully orientation equivariant architecture. Figure 2 exemplifies this issue: Figures 2a and 2b differ only in the orientation chosen to represent the scenario and, thus, any orientation equivariant model will output the same value for both. In Figure 2a, however, the flow orientation is defined opposite to the direction of a directed edge. In traffic applications, this drastically affects the scenario: In one case, one-way constraints are violated while in the other they are not.

This motivates our adopted notions of orientation equivariance and invariance: The concept of orientation is needed only for undirected edges. Directed edges already provide a reference orientation for representing signals through their direction. Furthermore, models should not be equivariant with respect to the direction of directed edges. In our work, we represent equivariant signals of directed edges with respect to their inherent direction and use an arbitrary orientation only for undirected edges. Consequently, the notions of orientation equivariance and orientation invariance only need to hold for undirected edges.

## C  THE LAPLACE OPERATORS OF EIGN

Here, we explicitly list the definitions of all Laplace operators and briefly describe some intuition. This is supplementary to the description for the Equivariant Edge Laplacian $L_{\text{equ}}$ in Section 4.3. Intuitively speaking, each Laplacian is a composition of two boundary operators. Therefore, the message passing from an adjacent edge $e' \in \mathcal{E}$ to a target edge $e \in \mathcal{E}$ can be understood as a two-step procedure: 1. The signal of $e'$ is expressed relative to the node both $e$ and $e'$ are incident to. This corresponds to the right-hand term in the construction of the Laplacian. Expressing the signal on the node level makes it invariant to the orientation (if the suitable boundary operator was used, i.e. the orientation-equivariant boundary for an orientation-equivariant signal and vice versa). 2. The signal of $e'$, currently expressed in reference to the shared node (ingoing versus outgoing), is then expressed relative to $e$. This corresponds to the left-hand term in the construction of the Laplacian from composing two boundaries. Depending on which boundary operator is used, this signal will be orientation-equivariant or orientation-invariant.

The values in Table 7 assume values based on if (i) $e$ and $e'$ align (i.e. are not consecutive). If they align, the first value is assumed, if they misalign, the second. (ii) The direction of $e$ relative to the node. For the direction depicted in Table 7, the first value is assumed. Flipping the orientation / direction of both edges gives the second value.

Based on this intuition, we interpret how the Laplacians of EIGN materialize in Table 7. We can see that only the Magnetic Laplacians are direction-aware and $L_{\text{equ}}$ and $L_{\text{inv}}$ do not depend on the direction. Furthermore, messages between two directed edges experience phase shifts twice: Once when being emitted from the directed source edge and once when being aggregated by the target edge. If the two directions are aligned, these phase shifts cancel out ($\xrightarrow{e}\xleftarrow{e'}$). Otherwise, they add to a total phase shift of $2\pi q$. One can also see that which direction an edge has (relative to the shared incident node) is encoded in the sign (i.e. direction) of the complex phase shift for both $L_{\text{equ}}^{(q)}$ and $L_{\text{inv}}^{(q)}$. The key difference between the orientation-equivariant $L_{\text{equ}}^{(q)}$ and the orientation-invariant $L_{\text{inv}}^{(q)}$ is that the former also re-orients signals depending on if the edges align, similar to $L_{\text{equ}}$. The Fusion Laplacians $L_{\text{equ} \to \text{inv}}^{(q)}$ and $L_{\text{inv} \to \text{equ}}^{(q)}$ mix both modalities and therefore can not be categorized intuitively as easily.

Since all Laplacians can be constructed without materializing any boundary operator by following Table 7, our approach scales in the number of edge pairs connected by a shared node. Put differently, EIGN can be seen as node-level GNN on the line graph of the problem with a sparse convolution

Table 7: Realization of different Laplacian operators for directed edges $\longrightarrow$ and undirected (oriented) edges $\Longrightarrow$. Equivariant operators encode if two edge orientations/directions align, i.e. whether they are non-consecutive through the sign of the real part. The sign of the complex phase determines the direction relative to the reference node both edges are incident to, i.e. whether they are in- or outgoing. The relative orientation (alignment) of two edges is ignored by orientation-invariant operators. We omit the cases for which the direction/orientation of both $e$ and $e'$ are flipped (this realizes a different sign in the complex phase).

| Adjacency | $[\boldsymbol{L}_{\text{equ}}]_{e,e'}$ | $[\boldsymbol{L}_{\text{equ}}^{(q)}]_{e,e'}$ | $[\boldsymbol{L}_{\text{inv}}]_{e,e'}$ | $[\boldsymbol{L}_{\text{inv}}^{(q)}]_{e,e'}$ | $[\boldsymbol{L}_{\text{equ}\to\text{inv}}^{(q)}]_{e,e'}$ | $[\boldsymbol{L}_{\text{inv}\to\text{equ}}^{(q)}]_{e,e'}$ |
|---|---|---|---|---|---|---|
| $e = e'$ | 2 | 2 | 2 | 2 | 2 | 2 |
| $\overset{e}{\longrightarrow}\overset{e'}{\longrightarrow}$ $\overset{e}{\longrightarrow}\overset{e'}{\longleftarrow}$ | $\pm 1$ | $\exp(\pm 2\pi q)$ $1$ | $1$ | $\exp(\pm 2\pi i q)$ $1$ | $\mp\exp(\pm 2\pi i q)$ $\pm 1$ | $\pm\exp(\pm 2\pi i q)$ $\pm 1$ |
| $\overset{e}{\longrightarrow}\overset{e'}{\Longrightarrow}$ $\overset{e}{\Longrightarrow}\overset{e'}{\longrightarrow}$ $\overset{e}{\longrightarrow}\overset{e'}{\Longleftarrow}$ $\overset{e}{\Longrightarrow}\overset{e'}{\longleftarrow}$ | $\pm 1$ | $\pm\exp(\pm\pi i q)$ | $1$ | $\exp(\pm\pi i q)$ | $\mp\exp(\pm\pi i q)$ $\pm\exp(\pm\pi i q)$ $\pm\exp(\mp\pi i q)$ | $\pm\exp(\pm\pi i q)$ $\pm\exp(\pm\pi i q)$ $\pm\exp(\mp\pi i q)$ |
| $\overset{e}{\Longrightarrow}\overset{e'}{\Longrightarrow}$ $\overset{e}{\Longrightarrow}\overset{e'}{\Longleftarrow}$ | $\pm 1$ | $\pm 1$ | $1$ | $1$ | $\mp 1$ $\pm 1$ | $\pm 1$ |

operator that has non-zero entries only when two of its nodes (i.e. edges) are adjacent. EIGN, therefore, has the same runtime complexity as all line-graph-like methods.

## D    EXPERIMENTAL DETAILS

### D.1    DATASETS

Here, we detail statistics for all datasets and the exact generation process for synthetic data. An overview of the statistics of each dataset is given in Table 8.

Table 8: Average performance of models on synthetic tasks (**best** and **runner-up**) with $95\%$ confidence intervals of the mean.

| Dataset | #Graphs | #Nodes | #Edges | | #Features | |
|---|---|---|---|---|---|---|
| | | | Dir. | Total | Equ. | Inv. |
| RW Comp | 1000 | 50 | 161–249 | 161–249 | 0 | 2 |
| LD Cycles | 1000 | 12–16 | 16–31 | 25–33 | 0 | 1 |
| Tri-Flow | 100 | 300 | 245–282 | 400 | 1 | 3 |
| Anaheim | 1 | 416 | 354 | 634 | 0 | 8 |
| Barcelona | 1 | 930 | 1074 | 1798 | 0 | 9 |
| Chicago | 1 | 933 | 0 | 1475 | 0 | 10 |
| Winnepeg | 1 | 1040 | 354 | 1595 | 0 | 8 |
| Circuits | 591 | 8–12 | 0–7 | 12–20 | 1 | 4 |

**RW Comp.** In this inductive setting, we generate $1,000$ Erdős–Rényi graphs Erdos et al. (1960) with $n = 50$ nodes and choose the edge probability such that the expected number of edges is 200. All edges are directed. Each graph is assigned a transition probability sampled uniformly from $[0, 1]$. For each graph, we sample a random walk starting from a randomly selected node (uniform probabilities) up to length 100. We then randomly select $20\%$ of the sampled transitions and provide them as orientation-invariant features as one-hot encoding and the transition probabilities. The task is to predict if a node is part of the $80\%$ nodes in the random walk that are not provided as an input, i.e. a binary classification problem. We use a $70/10/20$ split to assign graphs to train, validation, and test sets respectively.

**LD Cycles.** In this inductive setting, we generate $1,000$ graphs and sample a cycle size $c$ uniformly from $[6, 8]$. We then generate two cycles, each consisting of $c$ nodes: The first one contains only directed and consecutive edges. The second one is similar but contains one undirected edge. To each cycle, we then add additional $c$ random edges between its nodes, each of which is directed with a probability of $25\%$. We ensure that no second consecutive, purely directed cycle of length $c$ is manually formed in this process. Lastly, we connect both cycles with an edge that, too, is directed with a probability of $25\%$. Therefore, each graph can be seen as consisting of two components: One contains one consecutive, purely directed cycle of length $c$, while the other contains a non-purely directed, consecutive cycle of the same length. The task is to classify edges as being part of the longest purely-directed cycle or not. We provide the constant 1 vector as a surrogate orientation-invariant input, and, therefore, this task is purely topological. Models need to understand the concept of edge direction to perform well in this setting. Again, we use a $70/10/20$ split for the graphs.

**Tri-Flow.** In this inductive setting, we construct 100 graphs with $n = 300$ nodes and $m = 400$ edges. We want to induce a relationship between orientation-invariant and orientation-equivariant features. We assign each edge one of $c = 3$ colors as an orientation-invariant feature. We then plant $t = 100$ disjoint triangles in the graph. Each triangle is constructed according to one of three patterns: (i) Triangles with at least one directed edge, all edges of the same color. (ii) Triangles with edges of the same color, but at least two edges are directed and ensure that the triangle can not be consecutive. (iii) Triangles with edges of different colors. We generate $50\%$ of triangles with mechanism (i), and $25\%$ with mechanisms (ii) and (iii) respectively. We then add 100 random edges to connect the triangles and ensure that no new triangles are formed. For the task of this problem, we introduce an orientation-equivariant flow at each edge and the task is to re-orient these flow labels such that flows within triangles are closed. To that end, in the solution to this task, flow along directed edges must follow the edge direction while undirected edges permit flow in both directions. We introduce a dependency to orientation-invariant features by requiring that flow can only pass through triangles of the same color. That is, the target for triangles of type (i) is to re-orient the flow correctly such that it is closed within the monochromatic triangle. Triangles of type (ii) contain two directed edges that prevent consecutiveness and therefore, the target is 0 as no flow is permitted through such triangles. The same holds for triangles of type (iii) that prohibit flow as their edges are not monochromatic. We then add 100 edges, while ensuring no new triangles of type (i) are formed, that, consequentially, also have a target of 0. This task, therefore, involves understanding and relating the concepts of direction, orientation, color (which is orientation-invariant), and flow (which is orientation-equivariant). It probes all possible modalities an edge-level GNN can be exposed to. We again split each of the graphs into train, validation, and test sets using a $70/10/20$ split.

**Traffic Datasets.** We collect the Anaheim, Barcelona, Chicago, and Winnipeg datasets from the TNTP transportation network repository (Stabler et al., 2016). This project studies the Traffic Assignment Problem (Patriksson, 2015) and provides flows that correspond to the best-known solutions in terms of lowest Average Excess Cost (Boyce et al., 2004). We select these graphs based on that they are decently sized, provide both orientation-invariant features, and contain directed edges. Nodes correspond to intersections while edges represent links, e.g. streets. We use the following as orientation-invariant inputs (if available): (i) capacity, (ii) length, (iii) free flow time (i.e. travel time with no congestion), (iv) B factor and power, which are calibration parameters for the Traffic Assignment Problem, (v) if there is a toll on the link, and (vi) the link type, e.g. highway. (vii) if the edge corresponds to excess flow on a source / sink node The optimal flow solution is computed with respect to pre-defined demand for given source-target node pairs. To represent this as edge features, we identify all edges incident to edges that correspond to sources or sinks in terms of the transportation problem, add them to the training set and identify these edges with an invariant feature. By keeping these edges in the training set, the model can infer the problem constraints and predict flow on all edges that are not incident to sources or targets. The problem topology is preprocessed as follows: The input data is entirely directed, and, therefore, we aggregate edges $(u, v)$ and $(v, u)$ into one undirected edge if both are present. The flow labels are combined using subtraction (i.e. we subtract the from the edge direction that is used as a reference orientation to represent the flow). We then normalize the orientation-invariant features to follow a standard normal and normalize the target flows to $[0, 1]$.

**Electrical Circuits.** We generate random topologies with the following procedure: First, a cycle of length $c_{\text{init}} = 3$ is generated. Then, we iteratively attach new nodes $v$ to the graph by selecting random source and sink nodes $s, t$ and adding the edges $(s, v)$ and $(v, t)$. We then randomly assign different

component types to each edge: (i) exactly one power outlet, (ii) resistors with a resistance uniformly sampled from $[100\Omega, 10,000\Omega]$, (iii) diodes with saturation current $18.8nA$, parasitic resistance $0\Omega$, reverse breakdown voltage $0.5\mu V$, zero-bias junction capacitance $30F$, a linearly graded junction, emission coefficient of $2.0$ and transit time of $0s$. Diodes only permit current to flow in one direction. The ratio of resistors and diodes is $80/20$. We then use LTSpice (Asadi, 2022) to simulate the currents at each edge and voltages at each node until a steady state is reached. We use the component type and resistance at resistors as orientation-invariant features and normalize the latter to follow a standard normal distribution. The voltage along the power outlet is the only orientation-equivariant input signal, and we assign $0$ to all the other edges. For each graph we individually normalize the currents: First, we divide by the voltage at the power outlet. We then also normalize the currents again by dividing by the standard deviation of all flows on all graphs, i.e. a constant.

## D.2 TASKS

**Denoising.** The first task is denoising the flow signal of traffic data and circuits. To that end, for each graph, we compute the standard deviation $\sigma_y$ of all flows and then add noise sampled uniformly from $[-\sigma_y, \sigma_y]$ to all edges. The task is to recover the original signal, which we measure with the Root Mean Squared Error (RMSE) ($\downarrow$). Since the additional orientation-invariant input, the noisy flow provides a strong signal for the direction of the target, it is, arguably, considered to be the easiest of the three tasks.

**Interpolation.** The second task is predicting the traffic flow / current from only a few ground-truth values. To that end, for each graph, we provide $10\%$ of the true flow labels as an additional orientation-equivariant input and set the rest to $0$. The task is to predict the orientation-equivariant flow at edges for which flow was not provided. Again, we use RMSE as the main evaluation metric. Similar to the denoising task, the auxiliary orientation-invariant input signal can provide information about the true direction of the orientation-equivariant target. Since this information is now only available at some edges, we consider the interpolation task to be harder.

**Simulation.** The third task is predicting the orientation-equivariant target (traffic flow / electrical current) from just the topology and available orientation-equivariant and orientation-invariant features. This task can be seen as learning to simulate the traffic flow / electrical current. An effective model has to learn the underlying physical processes the data is derived from. Since we do not provide any input regarding the direction of the orientation-equivariant target, we consider this task to be the hardest problem among the three. Again, we can measure the performance of a model using RMSE but also report the Mean Absolute Error (MAE) ($\downarrow$) and $R^2$ value ($\uparrow$).

## D.3 MODEL ARCHITECTURE

**MLP.** We use a standard MLP with ReLU activations. We concatenate both orientation-equivariant and orientation-invariant inputs as a model input. This, effectively, mistreats orientation-equivariant input signals as orientation-invariant. The same is true for the orientation-equivariant model output/target.

**LINEGRAPH.** Similar to the MLP, we concatenate orientation-equivariant and orientation-invariant inputs, thereby mistreating the equivariant modality, and successively apply convolutions with the Line Graph Laplacian $\boldsymbol{L}_{\text{LINEGRAPH}} = \boldsymbol{D}_{\text{LINEGRAPH}} - \boldsymbol{A}_{\text{LINEGRAPH}}$:

$$[\boldsymbol{A}_{\text{LINEGRAPH}}]_{e,e'} = \begin{cases} 1 & \text{if } e \neq e' \text{ and } e, e' \text{ adjacent} \\ 0 & \text{otherwise} \end{cases}. \tag{10}$$

Here, $\boldsymbol{D}_{\text{LINEGRAPH}}$ is a diagonal matrix of edge degrees, i.e. $[\boldsymbol{D}_{\text{LINEGRAPH}}]_{e,e} = \sum_{e'}[\boldsymbol{A}_{\text{LINEGRAPH}}]_{e,e'}$. A LINEGRAPH convolution is then realized as:

$$\boldsymbol{H}^{(l)} = \sigma(\boldsymbol{L}_{\text{LINEGRAPH}}\boldsymbol{H}^{(l-1)}\boldsymbol{W}^{(l)} + \boldsymbol{b}^{(l)}). \tag{11}$$

Here, $\sigma$ is the ReLU non-linearity, and $\boldsymbol{W}^{(l)}$ and $\boldsymbol{b}^{(l)}$ are the trainable model parameters.

**HODGEGNN.** We use the HODGEGNN architecture of Roddenberry & Segarra (2019), which is a (jointly) orientation-equivariant architecture. It, however, can not model orientation-invariant inputs,

as it treats all input signals as equivariant. We therefore do not input this modality at all. Each layer uses the Equivariant Edge Laplacian $\boldsymbol{L}_{\text{equ}}$ as a convolution operator.

$$\boldsymbol{H}^{(l)} = \sigma_{\text{equ}}(\boldsymbol{L}_{\text{equ}}\boldsymbol{H}^{(l-1)}\boldsymbol{W}^{(l)}) \, . \tag{12}$$

Here, $\sigma_{\text{equ}}$ is a sign-equivariant activation function, which we initialize as $\tanh$. To not break (joint) orientation equivariance, we can not use a bias term.

**HODGE+INV.** We use the same architecture as for HODGEGNN, but also concatenate the orientation-invariant inputs to the orientation-equivariant inputs. This incorrectly models orientation-invariant signals as orientation-equivariant.

**HODGE+DIR.** We use the same architecture as for HODGEGNN, but use the ReLU as an activation function instead of the sign-equivariant $\tanh$. This breaks (joint) orientation equivariance: The model is now sensitive to the chosen orientation and, therefore, can be seen as direction-aware. However, this holds for all edges, and hence HODGE+DIR can also not model undirected edges anymore.

**LINE-MAGNET.** We adopt the node-level graph transformer of Geisler et al. (2023) for edge-level problems on directed graphs as follows: We first compute the spectral decomposition of the orientation equivariant and invariant Laplacians $\boldsymbol{L}_{\text{equ}}^{(q)}$ and $\boldsymbol{L}_{\text{inv}}^{(q)}$ as they are both normal matrices:

$$\boldsymbol{L}_{\text{equ}}^{(q)} = (\Gamma^{(q)})_{\text{equ}}^{H}\Lambda_{\text{equ}}^{(q)}\Gamma_{\text{equ}}^{(q)} \qquad\qquad \boldsymbol{L}_{\text{inv}}^{(q)} = (\Gamma^{(q)})_{\text{inv}}^{H}\Lambda_{\text{inv}}^{(q)}\Gamma_{\text{inv}}^{(q)} \, . \tag{13}$$

We then concatenate orientation equivariant and invariant edge features together with the $k = 32$ (complex-valued) eigenvectors $\Gamma_{\text{equ}}^{(q)}$ and $\Gamma_{\text{inv}}^{(q)}$ associated with the corresponding eigenvalues of smallest magnitude. We flatten real and complex parts. Similar to Geisler et al. (2023), these can be understood as direction-aware positional encodings of the edges in the graph. We then feed this as an input to a 4-layer transformer with hidden dimension $d = 32$, mirroring the hyperparameter choices for EIGN. Note that while this model can distinguish directed and undirected edges it is not a jointly orientation equivariant or invariant model according to Definitions 4.1 and 4.2 despite utilizing $\boldsymbol{L}_{\text{equ}}^{(q)}$ and $\boldsymbol{L}_{\text{inv}}^{(q)}$.

**Dir-GNN.** We also ablate an alternative mechanism to represent directed edges while correctly modeling orientation equivariant and invariant signals. The core idea closely follows the node-level GNN proposed by Rossi et al. (2023) and distinguishes directed from undirected edges through two different message-passing schemes. To that end, we separate the boundary operators $\boldsymbol{B}_{\text{equ}}$ and $\boldsymbol{B}_{\text{inv}}$ into three boundary maps each:

$$[\overleftrightarrow{B}_{\text{equ}}]_{v,e} = \begin{cases} -1 & \text{if } \mathfrak{O}(e) = (v, \cdot) \text{ and } e \in \mathcal{E}_{\text{U}} \\ 1 & \text{if } \mathfrak{O}(e) = (\cdot, v) \text{ and } e \in \mathcal{E}_{\text{U}} \\ 0 & \text{otherwise} \end{cases} \, . \tag{14}$$

$$[\overleftarrow{B}_{\text{equ}}]_{v,e} = \begin{cases} -1 & \text{if } e = (v, \cdot) \text{ and } e \in \mathcal{E}_{\text{D}} \\ 0 & \text{otherwise} \end{cases} \, . \tag{15}$$

$$[\overrightarrow{B}_{\text{equ}}]_{v,e} = \begin{cases} 1 & \text{if } e = (\cdot, v) \text{ and } e \in \mathcal{E}_{\text{D}} \\ 0 & \text{otherwise} \end{cases} \, . \tag{16}$$

$$[\overleftrightarrow{B}_{\text{inv}}]_{v,e} = \begin{cases} 1 & \text{if } v \in \mathfrak{O}(e) \text{ and } e \in \mathcal{E}_{\text{U}} \\ 0 & \text{otherwise} \end{cases} \, . \tag{17}$$

$$[\overleftarrow{B}_{\text{inv}}]_{v,e} = \begin{cases} 1 & \text{if } e = (v, \cdot) \text{ and } e \in \mathcal{E}_{\text{D}} \\ 0 & \text{otherwise} \end{cases} \, . \tag{18}$$

$$[\overrightarrow{B}_{\text{inv}}]_{v,e} = \begin{cases} 1 & \text{if } e = (\cdot, v) \text{ and } e \in \mathcal{E}_{\text{D}} \\ 0 & \text{otherwise} \end{cases} \, . \tag{19}$$

They imply three separate Edge Laplacians each set corresponding to one of the Magnetic operators proposed in our work through a similar construction. Instead of relying on complex numbers to represent directionality, this approach represents directed edges through three separate message-passing operations: One between undirected edges, one between directed edges that are both incoming

edges concerning a shared node, and one between edges that are outgoing concerning a shared node. One natural drawback of this approach is that there is no direct message passing between all three types of edges (like in EIGN). We construct the Dir-GNN architecture similar to EIGN: In Equations (2) and (3), we replace our Magnetic Laplacian convolutions with the aforementioned three Laplacians and associate a separate set of learnable weight matrices for each of the three. Summation serves to aggregate the result of the three distinct convolutions. All remaining design choices of EIGN remain. Therefore, this baseline closely ablates an alternative representation of edge directionality. Importantly, since the boundary operators are adapted from the ones proposed in our work, Dir-GNN, too, is a jointy orientation equivariant and invariant model according to Definitions 4.1 and 4.2.

**EIGN.** We realize EIGN according to Equations (2) to (5). We realize the sign-equivariant activation $\sigma_{\text{equ}}$ as $\tanh$ to ensure joint orientation equivariance (see Appendix A. In the main text, we omitted biases for brevity. However, to not violate joint orientation equivariance, biases can only be used when the output of the affine transformation is not orientation-equivariant, i.e. orientation-invariant. Consequently, the following linear transformations are supplied with biases: $\boldsymbol{W}_{\text{inv}\to\text{inv}}^{(l)}, \boldsymbol{W}_{\text{equ}\to\text{inv}}^{(l)}, \boldsymbol{W}_{\text{inv}}^{(l)}, \boldsymbol{W}_{\text{F,inv}\to\text{equ}}^{(l)}, \boldsymbol{W}_{\text{F,inv}\to\text{inv}}^{(l)}$. For each message passing operation, we realize the node transformation $h$ as a simple 1-layer MLP with hidden dimension 32 for each of the inter-modality convolutions $f_{\text{inv}\to\text{equ}}^{(l)}$ and $f_{\text{equ}\to\text{inv}}^{(l)}$, and as identities for the intra-modality convolutions $f_{\text{equ}\to\text{equ}}^{(l)}$ and $f_{\text{inv}\to\text{inv}}^{(l)}$ which we empirically find to perform better.

A second caveat is that EIGN's convolution operators output complex-valued signals. While one could, in general, backpropagate through such signals as well, we instead flatten the output signal of each convolution operation by concatenating real and imaginary parts. Notice that this operation is sign-invariant:

$$\text{FLATTEN}(-\boldsymbol{X}) = (-\operatorname{Re}(\boldsymbol{X})) \,\|\, (-\operatorname{Im}(\boldsymbol{X})) \tag{20}$$
$$= -(\operatorname{Re}(\boldsymbol{X}) \,\|\, \operatorname{Im}(\boldsymbol{X})) \tag{21}$$
$$= -\text{FLATTEN}(\boldsymbol{X}) \tag{22}$$

By Proposition A.4, applying this flattening operation, therefore, preserves orientation-equivariance. Orientation invariance is trivially preserved as there are no restrictions on the mappings applied to this signal modality as long as they are orientation-independent. This way, all model weights act on real-valued signals. Due to the concatenation of real and imaginary values, we keep hidden dimensions consistent by letting the affine / linear transformations project into a space of half the desired dimension. We fix the complex phase shift to $q = m^{-1}$, as this is the longest cycle length possible in any graph. Consequently, potential side-effects due to the cyclical nature of complex phase shifts are avoided.

Third, we apply a final linear/affine transformation to the representation of the last layer (for the respective signal modality) $\boldsymbol{H}_{(\cdot)}^{(L)}$. In the case of orientation-invariant outputs, we can use an affine transformation with biases as orientation-independent transformations do not affect joint orientation invariance. For orientation-equivariant outputs, we have to omit biases and use a linear transformation only. This linear transformation is a sign-equivariant function and, again, preserves joint orientation equivariance as per Proposition A.4.

Lastly, we also omit the normalization of the Laplacians in the main text. Each Laplacian $\boldsymbol{L}_{(\cdot)}^{(\cdot)} = (\boldsymbol{B}_{(\cdot)}^{(\cdot)})^H \boldsymbol{B}_{(\cdot)}^{(\cdot)}$ is normalized symmetrically by normalizing the corresponding boundary maps:

$$\tilde{\boldsymbol{B}}_{(\cdot)}^{(\cdot)} = \boldsymbol{B}_{(\cdot)}^{(\cdot)}(\boldsymbol{D}_{(\cdot)}^{(\cdot)})^{-0.5}. \tag{23}$$

Here, $[(\boldsymbol{D}_{(\cdot)}^{(\cdot)})^{-0.5}]_{e,e} = \sum_{e'}[\text{abs}(\boldsymbol{L}_{(\cdot)}^{(\cdot)})]_{e,e'}$ is a diagonal matrix of degrees of the Laplacian.

Since the normalization is independent of the orientation (due to the absolute value), it does not affect joint orientation equivariance or invariance.

### D.4 TRAINING SETUP

For all models, including the baselines, we perform an extensive hyperparameter search to select the most suitable configuration. To that end, we perform 20 different data splits for each hyperparameter configuration and compare the average performance to select the best hyperparameter configuration for each model. We search a cartesian grid of the following hyperparameter options:

- Learning Rate: $\{0.03, 0.01, 0.003, 0.001\}$
- Hidden Dimension: $\{8, 16, 32\}$
- Number of Layers: $\{2, 3, 4\}$

For the LD Cycles, we always fix the number of layers to the cycle size $c$ since our convolution operators are local and cycles can not be detected if the receptive field of an edge can not access the entire cycle.

In all reported results here, we compute average metrics over 50 different dataset splits and also provide the $95\%$ confidence interval for the mean.

All models are trained with the ADAM optimizer with no weight decay, a mean-squared error objective for regression problems, and cross-entropy for classification tasks. We clip gradients to norm 1 and apply dropout (Srivastava et al., 2014) with probability 0.1 after every layer. We use the following configurations for each dataset:

- Electrical Circuits: Batch size of 10 graphs, 200 epochs.
- Traffic Datasets: Batch size of 1, 500 epochs.
- RW Comp, LD Cycles: Batch size of 10 graphs, 50 epochs.
- Tri-Flow: Batch size of 1 graph, 50 epochs.

For the denoising, interpolation and simulation tasks we use a dataset split of $80/10/10$ for traffic and $50/25/25$ for the circuits dataset. For synthetic tasks, we use $70/10/20$. We track the validation RMSE over all epochs and select the model with the best validation metric. We implement our models in PyTorch (Paszke et al., 2017) and PyTorch Geometric (Fey & Lenssen, 2019) and train on these types of GPUs: (i) NVIDIA GTX 1080TI GPU (ii) NVIDIA A100 GPU (iii) NVIDIA H100 GPU .

## E ADDITIONAL RESULTS

Here, we report average results over 50 runs with the best-found hyperparameter configuration for every model. We also report the $95\%$ confidence interval for the mean.

**Synthetic Tasks and Simulation.** We supply the confidence intervals for tables Tables 4 and 5 in Tables 9 and 12. Additionally, we report Mean Absolute Error (MAE) ($\downarrow$) and $R^2$ score ($\uparrow$) for the simulation task in Tables 13 and 14. The findings are consistent with the RMSE metric.

**Denoising and Interpolation.** We show the RMSE for the denoising task in Table 10. EIGN achieves the best results on four out of five datasets. Because the noisy flow input makes this problem significantly easier, some baselines achieve satisfactory performance and the gap to EIGN is not as pronounced as in the challenging simulation task. We make similar observations for the interpolation problem in Table 11.

**Ablation.** We supply the confidence intervals for the mean RMSE of Table 6 in Table 15. Additionally, we visualize the distribution of simulated electrical currents (orientation-equivariant targets) in Figure 6. Similar to Figure 5, we find that the direction-aware EIGN provides physically more plausible predictions. The non-negativity constraint imposed by diodes is learned to a lesser extent than on the Anaheim dataset (Figure 5).

**Phase shift $q$.** Figure 7 showcases the performance of EIGN and two variants without inter-modality convolutions and fusion respectively for different relative phase shifts $q/m$. EIGN suffers from phase shifts that are picked too small as such shifts become too small to notice. In contrast, if the phase shift is picked too large, accumulated phase shifts may overshoot $2\pi$. For example, choosing $q = 2\pi k$ for $k \in \mathbb{N}$ is equivalent to applying a phase shift of 0, i.e. no phase shift. However, even

Table 9: Average performance of models on synthetic tasks (**best** and **runner-up**) with $95\%$ confidence intervals of the mean.

| Model | RW Comp AUC-ROC($\uparrow$) | LD Cycles AUC-ROC($\uparrow$) | Tri-Flow RMSE($\downarrow$) |
|---|---|---|---|
| MLP | $0.720_{\pm 0.001}$ | $0.500_{\pm 0.000}$ | $0.547_{\pm 0.001}$ |
| LINEGRAPH | $0.758_{\pm 0.001}$ | $0.683_{\pm 0.001}$ | $0.497_{\pm 0.002}$ |
| HODGEGNN | $0.500_{\pm 0.000}$ | $0.500_{\pm 0.000}$ | $0.458_{\pm 0.001}$ |
| HODGE+INV | $0.811_{\pm 0.001}$ | $0.754_{\pm 0.005}$ | $\mathbf{0.293_{\pm 0.002}}$ |
| HODGE+DIR | $\mathbf{0.819_{\pm 0.001}}$ | $\mathbf{0.799_{\pm 0.008}}$ | $\mathbf{0.293_{\pm 0.003}}$ |
| LINE-MAGNET | $0.729_{\pm 0.001}$ | $0.502_{\pm 0.002}$ | $0.542_{\pm 0.001}$ |
| Dir-GNN | $0.757_{\pm 0.001}$ | $0.768_{\pm 0.004}$ | $0.453_{\pm 0.002}$ |
| EIGN | $\mathbf{0.864_{\pm 0.001}}$ | $\mathbf{0.996_{\pm 0.001}}$ | $\mathbf{0.022_{\pm 0.002}}$ |

Table 10: Average RMSE ($\downarrow$) of different for the denoising task on real-world datasets (**best** and **runner-up**) with $95\%$ confidence intervals of the mean.

| Model | Anaheim | Barcelona | Chicago | Winnipeg | Circuits |
|---|---|---|---|---|---|
| MLP | $0.076_{\pm 0.001}$ | $0.063_{\pm 0.001}$ | $\mathbf{0.028_{\pm 0.000}}$ | $0.055_{\pm 0.000}$ | $0.462_{\pm 0.009}$ |
| LINEGRAPH | $0.070_{\pm 0.001}$ | $0.059_{\pm 0.000}$ | $0.030_{\pm 0.000}$ | $0.054_{\pm 0.000}$ | $0.443_{\pm 0.013}$ |
| HODGEGNN | $0.084_{\pm 0.001}$ | $0.063_{\pm 0.000}$ | $0.048_{\pm 0.001}$ | $0.069_{\pm 0.000}$ | $0.360_{\pm 0.007}$ |
| HODGE+INV | $0.062_{\pm 0.001}$ | $0.056_{\pm 0.000}$ | $0.030_{\pm 0.000}$ | $0.051_{\pm 0.000}$ | $0.346_{\pm 0.011}$ |
| HODGE+DIR | $\mathbf{0.053_{\pm 0.001}}$ | $\mathbf{0.048_{\pm 0.000}}$ | $\mathbf{0.025_{\pm 0.000}}$ | $\mathbf{0.041_{\pm 0.000}}$ | $\mathbf{0.262_{\pm 0.009}}$ |
| LINE-MAGNET | $0.096_{\pm 0.001}$ | $0.066_{\pm 0.000}$ | $0.034_{\pm 0.000}$ | $0.062_{\pm 0.000}$ | $0.508_{\pm 0.011}$ |
| Dir-GNN | $0.069_{\pm 0.001}$ | $0.066_{\pm 0.001}$ | $0.036_{\pm 0.000}$ | $0.061_{\pm 0.001}$ | $0.422_{\pm 0.020}$ |
| EIGN | $\mathbf{0.050_{\pm 0.001}}$ | $\mathbf{0.053_{\pm 0.000}}$ | $0.039_{\pm 0.001}$ | $\mathbf{0.050_{\pm 0.001}}$ | $\mathbf{0.266_{\pm 0.017}}$ |

smaller values of $q$ may cause problems as discussed in (Geisler et al., 2023): Consider any pair of directed edges connected through a sequence of $L$ other directed edges (e.g. in a cycle). After repeatedly applying the Magnetic Laplacian operators we propose the relative phase shift of these two edges will be $L * q$. Since these relative phases are, however, taken modulo $2\pi$ this introduces ambiguities if $L * q$ exceeds $2\pi$.

To mitigate this issue, we choose $q = 1/m$ which ensures such issues can not arise. In practice, this may be a conservative value and we, in fact, also observe in Figure 7 that slightly larger values improve performance in interpolation and simulation problems. Interestingly, the pattern slightly deviates for the easier denoising task: We conjecture that there identifying directed edges may not play as large of a role as much of the information is already encoded in the noisy input signal.

**Hidden Size and Number of Layers.** Table 16 ablates different hyperparamter configurations of EIGN on the electrical circuits simulation task. In particular, we vary the learning rate, number of hidden dimensions and number of layers as described in Appendix D.4. In general, we observe that both under-parametrized models (i.e. small number of layers and / or hidden dimension) and too larger learning rate lead to worse performance. However, models that are sufficiently deep and wide enough are consistently able to outperform the baselines in terms of RMSE.

### E.1 VARIATIONS ON EIGN AND BASELINES

**GCN-like Convolutions.** Instead of using our proposed (Magnetic) Laplacians as graph-shift operators according to Equations (2) and (3), we also ablate an operator $\boldsymbol{A}_{(\cdot)}^{(\cdot)}$ that is defined as $\boldsymbol{A}_{(\cdot)}^{(\cdot)} = \boldsymbol{I} - \boldsymbol{L}_{(\cdot)}^{(\cdot)}/2$ akin to GCNs Kipf & Welling (2017) in Table 17 In particular, we replace each Laplacian with its corresponding GCN-like counterpart $\boldsymbol{A}_{(\cdot)}^{(\cdot)}$ and leave all other components of EIGN unaffected.

**Chebyshev Convolutions.** Similarly, following a recent line of work on using polynomials of the Node Laplacian as graph-shift operator Defferrard et al. (2016). In particular, for each Magnetic Laplacian operator $\boldsymbol{L}_{(\cdot)}^{(q)}$ we instead convole an input signal $\boldsymbol{H}_{(\cdot)}$ of each respective modality with a

Table 11: Average RMSE ($\downarrow$) of different for the interpolation task on real-world datasets (**best** and **runner-up**) with $95\%$ confidence intervals of the mean.

| Model | Anaheim | Barcelona | Chicago | Winnipeg | Circuits |
|---|---|---|---|---|---|
| MLP | $0.103_{\pm 0.001}$ | $0.150_{\pm 0.001}$ | $0.110_{\pm 0.002}$ | $0.166_{\pm 0.001}$ | $1.030_{\pm 0.034}$ |
| LINEGRAPH | $0.103_{\pm 0.001}$ | $0.148_{\pm 0.002}$ | $0.109_{\pm 0.002}$ | $0.165_{\pm 0.001}$ | $1.025_{\pm 0.032}$ |
| HODGEGNN | $0.250_{\pm 0.003}$ | $0.166_{\pm 0.001}$ | $\mathbf{0.105}_{\pm 0.001}$ | $0.166_{\pm 0.001}$ | $0.997_{\pm 0.034}$ |
| HODGE+INV | $0.095_{\pm 0.001}$ | $0.144_{\pm 0.001}$ | $0.108_{\pm 0.001}$ | $0.145_{\pm 0.001}$ | $0.778_{\pm 0.026}$ |
| HODGE+DIR | $\mathbf{0.088}_{\pm 0.001}$ | $\mathbf{0.141}_{\pm 0.002}$ | $0.112_{\pm 0.002}$ | $\mathbf{0.133}_{\pm 0.001}$ | $\mathbf{0.753}_{\pm 0.029}$ |
| LINE-MAGNET | $0.116_{\pm 0.001}$ | $0.151_{\pm 0.002}$ | $0.106_{\pm 0.001}$ | $0.171_{\pm 0.001}$ | $1.031_{\pm 0.033}$ |
| Dir-GNN | $0.277_{\pm 0.003}$ | $0.169_{\pm 0.002}$ | $0.109_{\pm 0.002}$ | $0.160_{\pm 0.002}$ | $0.945_{\pm 0.026}$ |
| EIGN | $\mathbf{0.081}_{\pm 0.002}$ | $\mathbf{0.125}_{\pm 0.002}$ | $\mathbf{0.081}_{\pm 0.001}$ | $\mathbf{0.100}_{\pm 0.001}$ | $\mathbf{0.600}_{\pm 0.032}$ |

Table 12: Average RMSE ($\downarrow$) of different for the simulation task on real-world datasets (**best** and **runner-up**) with $95\%$ confidence intervals of the mean.

| Model | Anaheim | Barcelona | Chicago | Winnipeg | Circuits |
|---|---|---|---|---|---|
| MLP | $0.105_{\pm 0.001}$ | $0.149_{\pm 0.001}$ | $0.109_{\pm 0.002}$ | $0.167_{\pm 0.001}$ | $1.030_{\pm 0.035}$ |
| LINEGRAPH | $0.101_{\pm 0.001}$ | $0.149_{\pm 0.002}$ | $0.109_{\pm 0.002}$ | $0.164_{\pm 0.002}$ | $1.037_{\pm 0.038}$ |
| HODGEGNN | $0.280_{\pm 0.003}$ | $0.170_{\pm 0.002}$ | $0.107_{\pm 0.001}$ | $0.173_{\pm 0.001}$ | $1.016_{\pm 0.034}$ |
| HODGE+INV | $0.098_{\pm 0.001}$ | $0.146_{\pm 0.001}$ | $0.108_{\pm 0.001}$ | $0.151_{\pm 0.001}$ | $0.828_{\pm 0.026}$ |
| HODGE+DIR | $\mathbf{0.091}_{\pm 0.001}$ | $\mathbf{0.144}_{\pm 0.002}$ | $0.109_{\pm 0.001}$ | $\mathbf{0.132}_{\pm 0.001}$ | $\mathbf{0.760}_{\pm 0.030}$ |
| LINE-MAGNET | $0.119_{\pm 0.002}$ | $0.151_{\pm 0.002}$ | $\mathbf{0.105}_{\pm 0.001}$ | $0.170_{\pm 0.001}$ | $1.027_{\pm 0.032}$ |
| Dir-GNN | $0.278_{\pm 0.002}$ | $0.170_{\pm 0.001}$ | $0.106_{\pm 0.001}$ | $0.173_{\pm 0.001}$ | $1.029_{\pm 0.033}$ |
| EIGN | $\mathbf{0.090}_{\pm 0.002}$ | $\mathbf{0.133}_{\pm 0.002}$ | $\mathbf{0.078}_{\pm 0.001}$ | $\mathbf{0.101}_{\pm 0.001}$ | $\mathbf{0.696}_{\pm 0.038}$ |

Chebyshev polynomial of order $k = 5$. The resulting filter can be defined recursively as follows:

$$\boldsymbol{C}_{(.)}^{(1)} = \boldsymbol{H}_{(.)} \tag{24}$$

$$\boldsymbol{C}_{(.)}^{(2)} = \boldsymbol{L}_{(.)}^{(q)} \boldsymbol{H}_{(.)} \tag{25}$$

$$\boldsymbol{C}_{(.)}^{(k)} = 2\hat{\boldsymbol{L}}_{(.)}^{(q)} \boldsymbol{C}_{(.)}^{(k-1)} - \boldsymbol{C}_{(.)}^{(k-2)} \tag{26}$$

For the convolution operators within a signal modality $\boldsymbol{L}_{\text{equ}}^{(q)}$ and $\boldsymbol{L}_{\text{inv}}^{(q)}$, we set $\hat{\boldsymbol{L}}_{\text{equ}}^{(q)} = \boldsymbol{L}_{\text{equ}}^{(q)}$ and $\hat{\boldsymbol{L}}_{\text{inv}}^{(q)} = \boldsymbol{L}_{\text{inv}}^{(q)}$. For the inter-modality Laplacians, however, using higher powers would break joint orientation equivariance and invariance respectively. That is, because the operators transform one signal modality into the other and repeated application would apply the wrong Laplacian to different modalities. We therefore only apply the inter-modality Laplacians for the first term $\boldsymbol{C}_{(.)}^{(2)}$ and use the Laplacians of the target modality for higher order terms, i.e. we set $\hat{\boldsymbol{L}}_{\text{equ}\to\text{inv}}^{(q)} = \boldsymbol{L}_{\text{inv}}^{(q)}$ and $\hat{\boldsymbol{L}}_{\text{inv}\to\text{equ}}^{(q)} = \boldsymbol{L}_{\text{equ}}^{(q)}$ respectively.

**Large Baselines.** While the search space over which we optimize the hyperparameters for each baseline is shared different architectures can result in models with different parameter counts even for the same hyperparameter settings. To that end, we evaluate our baselines for a hidden dimension of $d = 80$ for four layers. These models roughly have the same number of parameters as EIGN. In Table 17, we find that even for comparable parameter counts EIGN outperforms the competitors. This underlines that it is the inductive biases that carefully follow from our theoretical considerations which are responsible for the high efficacy of EIGN and not its larger number of parameters.

Table 13: Average MAE ($\downarrow$) of different for the simulation task on real-world datasets (**best** and **runner-up**) with $95\%$ confidence intervals of the mean.

| Model | Anaheim | Barcelona | Chicago | Winnipeg | Circuits |
|---|---|---|---|---|---|
| MLP | $0.069_{\pm0.001}$ | $0.098_{\pm0.001}$ | $0.066_{\pm0.001}$ | $0.106_{\pm0.001}$ | $0.542_{\pm0.014}$ |
| LINEGRAPH | $0.068_{\pm0.001}$ | $0.095_{\pm0.001}$ | $0.066_{\pm0.001}$ | $0.104_{\pm0.001}$ | $0.549_{\pm0.014}$ |
| HODGEGNN | $0.183_{\pm0.002}$ | $0.102_{\pm0.001}$ | $\mathbf{0.065}_{\pm0.001}$ | $0.109_{\pm0.001}$ | $0.514_{\pm0.013}$ |
| HODGE+INV | $0.067_{\pm0.001}$ | $0.098_{\pm0.001}$ | $0.067_{\pm0.001}$ | $0.100_{\pm0.001}$ | $0.459_{\pm0.010}$ |
| HODGE+DIR | $\mathbf{0.062}_{\pm0.001}$ | $\mathbf{0.094}_{\pm0.001}$ | $0.066_{\pm0.001}$ | $\mathbf{0.089}_{\pm0.001}$ | $\mathbf{0.420}_{\pm0.011}$ |
| LINE-MAGNET | $0.079_{\pm0.001}$ | $0.097_{\pm0.001}$ | $\mathbf{0.065}_{\pm0.001}$ | $0.108_{\pm0.001}$ | $0.536_{\pm0.012}$ |
| Dir-GNN | $0.181_{\pm0.002}$ | $0.103_{\pm0.001}$ | $\mathbf{0.065}_{\pm0.001}$ | $0.109_{\pm0.001}$ | $0.518_{\pm0.013}$ |
| EIGN | $\mathbf{0.061}_{\pm0.001}$ | $\mathbf{0.088}_{\pm0.001}$ | $\mathbf{0.049}_{\pm0.001}$ | $\mathbf{0.069}_{\pm0.001}$ | $\mathbf{0.373}_{\pm0.014}$ |

Table 14: Average $R^2$ ($\uparrow$) of different for the simulation task on real-world datasets (**best** and **runner-up**) with $95\%$ confidence intervals of the mean.

| Model | Anaheim | Barcelona | Chicago | Winnipeg | Circuits |
|---|---|---|---|---|---|
| MLP | $0.890_{\pm0.003}$ | $0.298_{\pm0.008}$ | $0.003_{\pm0.015}$ | $0.216_{\pm0.009}$ | $0.116_{\pm0.011}$ |
| LINEGRAPH | $0.897_{\pm0.002}$ | $0.319_{\pm0.012}$ | n.a. | n.a. | $0.097_{\pm0.015}$ |
| HODGEGNN | $0.000_{\pm0.000}$ | $0.000_{\pm0.000}$ | $0.000_{\pm0.000}$ | $0.000_{\pm0.000}$ | $-0.039_{\pm0.014}$ |
| HODGE+INV | $0.905_{\pm0.002}$ | $0.339_{\pm0.006}$ | $-0.025_{\pm0.009}$ | $0.444_{\pm0.008}$ | $0.599_{\pm0.015}$ |
| HODGE+DIR | $\mathbf{0.919}_{\pm0.002}$ | $\mathbf{0.402}_{\pm0.010}$ | $-0.033_{\pm0.009}$ | $\mathbf{0.640}_{\pm0.008}$ | $\mathbf{0.656}_{\pm0.011}$ |
| LINE-MAGNET | $0.855_{\pm0.004}$ | $0.347_{\pm0.011}$ | $\mathbf{0.006}_{\pm0.008}$ | $0.213_{\pm0.009}$ | $0.090_{\pm0.011}$ |
| Dir-GNN | $0.000_{\pm0.000}$ | $0.000_{\pm0.000}$ | $0.000_{\pm0.000}$ | $0.000_{\pm0.000}$ | $-0.014_{\pm0.023}$ |
| EIGN | $\mathbf{0.917}_{\pm0.003}$ | $\mathbf{0.535}_{\pm0.013}$ | $\mathbf{0.679}_{\pm0.010}$ | $\mathbf{0.806}_{\pm0.005}$ | $\mathbf{0.744}_{\pm0.019}$ |

Table 15: Ablation of different components of EIGN on synthetic tasks and simulation on real data (**best** and **runner-up**) with $95\%$ confidence intervals of the mean. We omit (i) direction-awareness by setting $q = 0$, (ii) the fusion operation of Equations (4) and (5), (iii) the fusion operators using convolutions $\boldsymbol{L}^{(q)}_{\text{equ}\rightarrow\text{inv}}$ and $\boldsymbol{L}^{(q)}_{\text{inv}\rightarrow\text{equ}}$

| | Dataset | EIGN w/o Direction | EIGN No Fusion | EIGN No Fusion-Conv. | EIGN No $h$ | EIGN |
|---|---|---|---|---|---|---|
| AUC $\uparrow$ | RW Comp | $0.762_{\pm0.001}$ | $0.853_{\pm0.001}$ | $0.845_{\pm0.001}$ | $\mathbf{0.862}_{\pm0.001}$ | $\mathbf{0.864}_{\pm0.001}$ |
| | LD Cycles | $0.689_{\pm0.002}$ | $\mathbf{0.987}_{\pm0.001}$ | $0.926_{\pm0.003}$ | $\mathbf{0.996}_{\pm0.000}$ | $\mathbf{0.996}_{\pm0.001}$ |
| RMSE $\downarrow$ | Tri-Flow | $0.362_{\pm0.002}$ | $0.088_{\pm0.002}$ | $0.074_{\pm0.007}$ | $\mathbf{0.034}_{\pm0.003}$ | $\mathbf{0.022}_{\pm0.002}$ |
| | Anaheim | $0.289_{\pm0.016}$ | $\mathbf{0.097}_{\pm0.004}$ | $0.283_{\pm0.009}$ | $0.099_{\pm0.005}$ | $\mathbf{0.090}_{\pm0.002}$ |
| | Barcelona | $0.172_{\pm0.004}$ | $\mathbf{0.139}_{\pm0.005}$ | $0.177_{\pm0.004}$ | $0.163_{\pm0.007}$ | $\mathbf{0.133}_{\pm0.002}$ |
| | Chicago | $\mathbf{0.079}_{\pm0.004}$ | $0.093_{\pm0.003}$ | $0.110_{\pm0.006}$ | $0.082_{\pm0.005}$ | $\mathbf{0.078}_{\pm0.001}$ |
| | Winnipeg | $\mathbf{0.132}_{\pm0.005}$ | $0.170_{\pm0.005}$ | $0.175_{\pm0.004}$ | $0.138_{\pm0.004}$ | $\mathbf{0.101}_{\pm0.001}$ |
| | Circuits | $0.957_{\pm0.042}$ | $0.974_{\pm0.043}$ | $0.727_{\pm0.028}$ | $\mathbf{0.707}_{\pm0.035}$ | $\mathbf{0.696}_{\pm0.038}$ |

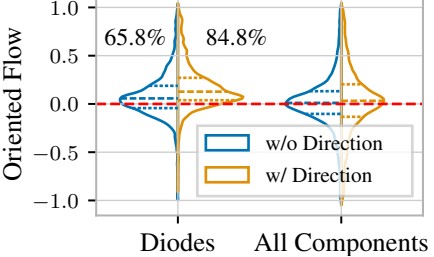

Figure 6: Distribution of simulated traffic on the electrical circuits dataset for EIGN and a direction-agnostic variant that sets q = 0. Direction-awareness leads to more more plausible predictions, i.e. current that follows the constraints of diode components that only permit one flow direction.

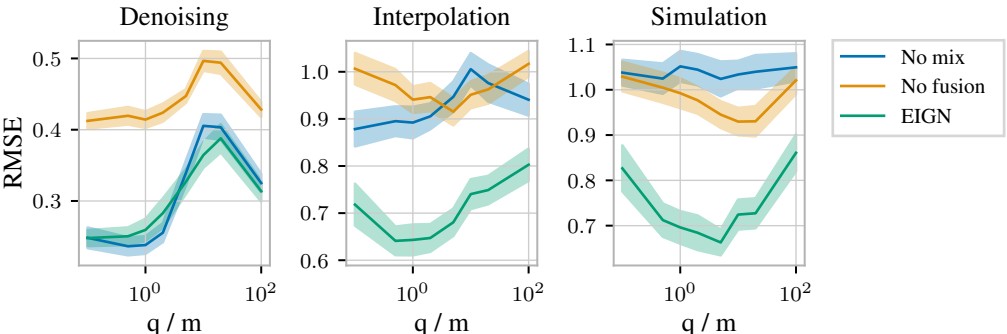

Figure 7: Ablation of the relative phase shift strength $q * m$ on the performance for all three tasks on the Circuits dataset. Phase shifts that are too small become hard to notice while too large phase shifts may accumulate to total phase shifts larger than $2\pi$.

Table 16: Average RMSE ($\downarrow$) of the simluation task on the circuits dataset for EIGN at different hyperparameter configurations with $95\%$ confidence intervals of the mean. **Bold** numbers indicate improvements over all baselines. We vary the learning rate (LR), hidden dimension $d$ and number of layers.

| LR | Hidden Dim. | 2-layers | 3-layers | 4-layers |
|---|---|---|---|---|
| 0.001 | 8 | $1.003_{\pm 0.033}$ | $0.948_{\pm 0.040}$ | $0.799_{\pm 0.036}$ |
| 0.001 | 16 | $0.974_{\pm 0.039}$ | $0.917_{\pm 0.044}$ | $0.810_{\pm 0.044}$ |
| 0.001 | 32 | $0.983_{\pm 0.037}$ | $0.931_{\pm 0.034}$ | $\mathbf{0.746_{\pm 0.042}}$ |
| 0.003 | 8 | $0.977_{\pm 0.037}$ | $0.827_{\pm 0.034}$ | $\mathbf{0.707_{\pm 0.033}}$ |
| 0.003 | 16 | $1.002_{\pm 0.036}$ | $0.834_{\pm 0.038}$ | $\mathbf{0.701_{\pm 0.035}}$ |
| 0.003 | 32 | $0.993_{\pm 0.030}$ | $0.837_{\pm 0.034}$ | $\mathbf{0.671_{\pm 0.033}}$ |
| 0.01 | 8 | $0.944_{\pm 0.040}$ | $0.769_{\pm 0.027}$ | $\mathbf{0.675_{\pm 0.031}}$ |
| 0.01 | 16 | $0.965_{\pm 0.035}$ | $0.776_{\pm 0.045}$ | $\mathbf{0.716_{\pm 0.030}}$ |
| 0.01 | 32 | $0.986_{\pm 0.031}$ | $0.828_{\pm 0.040}$ | $0.896_{\pm 0.039}$ |
| 0.03 | 8 | $0.909_{\pm 0.033}$ | $0.939_{\pm 0.032}$ | $0.966_{\pm 0.033}$ |
| 0.03 | 16 | $0.971_{\pm 0.024}$ | $0.994_{\pm 0.032}$ | $1.023_{\pm 0.033}$ |
| 0.03 | 32 | $0.968_{\pm 0.035}$ | $1.016_{\pm 0.033}$ | $0.994_{\pm 0.031}$ |

Table 17: Average RMSE ($\downarrow$) of different for the simulation task on real-world datasets (**best** and **runner-up**) with $95\%$ confidence intervals of the mean for baselines with paramater counts comparable to EIGN as well as variations on EIGN.

| Model | # params | Anaheim | Barcelona | Chicago | Winnipeg | Circuits |
|---|---|---|---|---|---|---|
| MLP | 209 | $0.105_{\pm 0.001}$ | $0.149_{\pm 0.001}$ | $0.109_{\pm 0.002}$ | $0.167_{\pm 0.001}$ | $1.030_{\pm 0.035}$ |
| LINEGRAPH | 753 | $0.101_{\pm 0.001}$ | $0.149_{\pm 0.002}$ | $0.109_{\pm 0.002}$ | $0.164_{\pm 0.002}$ | $1.037_{\pm 0.038}$ |
| HODGEGNN | 2k | $0.280_{\pm 0.003}$ | $0.170_{\pm 0.001}$ | $0.107_{\pm 0.001}$ | $0.173_{\pm 0.001}$ | $1.016_{\pm 0.034}$ |
| HODGE+INV | 2k | $0.098_{\pm 0.001}$ | $0.146_{\pm 0.001}$ | $0.108_{\pm 0.001}$ | $0.151_{\pm 0.001}$ | $0.828_{\pm 0.026}$ |
| HODGE+DIR | 2k | $0.091_{\pm 0.001}$ | $\mathbf{0.144}_{\pm 0.002}$ | $0.109_{\pm 0.001}$ | $\mathbf{0.132}_{\pm 0.001}$ | $0.760_{\pm 0.030}$ |
| LINE-MAGNET | 554k | $0.119_{\pm 0.002}$ | $0.151_{\pm 0.002}$ | $0.105_{\pm 0.001}$ | $0.170_{\pm 0.001}$ | $1.027_{\pm 0.032}$ |
| Dir-GNN | 42k | $0.278_{\pm 0.002}$ | $0.170_{\pm 0.001}$ | $0.106_{\pm 0.001}$ | $0.173_{\pm 0.001}$ | $1.029_{\pm 0.033}$ |
| MLP-L | 20k | $0.109_{\pm 0.005}$ | $0.146_{\pm 0.004}$ | $0.115_{\pm 0.006}$ | $0.166_{\pm 0.005}$ | $1.008_{\pm 0.029}$ |
| LINEGRAPH -L | 40k | $0.103_{\pm 0.005}$ | $0.152_{\pm 0.006}$ | $0.112_{\pm 0.007}$ | $0.163_{\pm 0.005}$ | $0.994_{\pm 0.036}$ |
| HODGEGNN-L | 39k | $0.279_{\pm 0.007}$ | $0.172_{\pm 0.004}$ | $0.109_{\pm 0.005}$ | $0.173_{\pm 0.004}$ | $1.035_{\pm 0.036}$ |
| HODGE+INV-L | 39k | $0.099_{\pm 0.004}$ | $0.146_{\pm 0.004}$ | $0.109_{\pm 0.006}$ | $0.157_{\pm 0.005}$ | $0.847_{\pm 0.030}$ |
| HODGE+DIR-L | 40k | $0.098_{\pm 0.004}$ | $0.152_{\pm 0.006}$ | $0.115_{\pm 0.006}$ | $0.135_{\pm 0.004}$ | $0.792_{\pm 0.028}$ |
| EIGN-GCN | 36k | $0.104_{\pm 0.004}$ | $0.146_{\pm 0.007}$ | $0.111_{\pm 0.006}$ | $0.139_{\pm 0.004}$ | $0.869_{\pm 0.034}$ |
| EIGN-Cheb | 134k | $\mathbf{0.078}_{\pm 0.006}$ | $0.159_{\pm 0.007}$ | $\mathbf{0.068}_{\pm 0.004}$ | $\mathbf{0.101}_{\pm 0.005}$ | $\mathbf{0.705}_{\pm 0.037}$ |
| EIGN | 36k | $\mathbf{0.090}_{\pm 0.002}$ | $\mathbf{0.133}_{\pm 0.002}$ | $\mathbf{0.078}_{\pm 0.001}$ | $\mathbf{0.101}_{\pm 0.001}$ | $\mathbf{0.696}_{\pm 0.038}$ |

