# OpenReview forum: "Graph Neural Networks for Edge Signals: Orientation Equivariance and Invariance"
_ICLR.cc/2025/Conference — ICLR 2025 Poster_

### Official Review · Reviewer_fGAT · 2024-10-31

**Soundness:** 4
**Presentation:** 4
**Contribution:** 4
**Rating:** 8
**Confidence:** 3

**Summary:**

This paper seeks to extend equivariant machine learning to signals (functions) defined on the vertices of a graph, including signals which are inherently directed (as well as undirected signals). It aims to produce representations which are either invariant or equivariant to changes in edge direction as appropriate, which is non-trivial since standard methods require a boundary operator which chooses an arbitrary edge direction.

It achieves its goals by formulating the notion of direction consistent orientations and uses this notation to formulate joint orientation equivariance / invariance and then applying several distinct convolution operations (with various generalized magnetic Laplacians ) together with a fusion module and showcases its numerical effectiveness on real and synthetic data sets including data sets from traffic, which was one of the motivating applications.

Overall: I am rating this paper a six, but would have rated this paper a seven if that were an option.

**Strengths:**

The tasks are well motivated by realworld scenarios – figure 2 is quite nice!

The proposed network is an elegant solution for achieving the desired properties

Experiments are well sought out and feature a thorough ablation

**Weaknesses:**

The use of separate equivariant/invariant Laplacains L_equ, L_{inv-> equ}, etc is quite interesting. However, it is unclear how the proposed model compares to other networks defined in these Lapacians. Could one instead constructed a ChebNet style architecture as was done in (Zhang et al 2021) with the magnetic Laplacian

I am unsure if the baselines adequately capture the current SotA


Minor:

Many equations, e.g., (6) and (7) are missing punctuation marks. (This does not affect my score but should be fixed)

The notation B^H is used without definition. I understand that this is the Hermitian transpose, but it should still be defined. (This does not affect my score but should be fixed)

Capitilization in the references e.g., ``laplacian” vs ``Laplacian”, should be fixed (This does not affect my score but should be fixed)

**Questions:**

The condition that \sigma_{equ}(-x)=-\simga_{\equ}(x) (line 242) seems stange. Are there common activation functions that satisfy this?

Why is there multiplication by the Laplacians in (2) and (3)? Based on standard GCNs it seems that left multiplication by low-pass filters (I-L/2) would be more natural?

Can this framework be extended to settings where there are both edge signals and node-signals?

Similarly, can it be extended to signed and directed graphs? The reference Zhang et al (2021), which the authors take setting from was extended to this setting in subsequent work ``MSGNN: A Spectral Graph Neural Network Based on a Novel Magnetic Signed Laplacian” (He et al.)

---

> ### Author Response · Authors · 2024-11-21
>
> We thank the reviewer for their thorough review and are happy that they find our work to be well-motivated and our approach to be elegant and thoroughly evaluated.
>
> ## Weakness 1, Question 2: Other Variations using the Novel Laplacians
>
> There are several ways to use our novel Laplacians for convolutions. Since the main focus of our paper lies on properly handling both signal modalities and the novel Laplace operators, we aimed not to overcomplicate the propagation scheme. Doing so would, for example, leave it open whether the success comes from our analysis or the more complex framework. We believe the success of using Laplacians as convolution operators makes a strong point in favor of our claim: By adequately treating different modalities a huge performance boost can be achieved already.
>
> As asked by the reviewer, we extend our evaluation by a ChebNet-style [3] model and a GCN-like convolution $I - L/2$ in Table 17. The Chebyshev polynomials we use are detailed in Appendix E.1. The GCN-like baselines performs sub-par, potentially because the alternative convolution operator interferes with complex phase shifts used by EIGN to encode direction. The ChebNet variant performs exceptionally well. It even outperforms the basic EIGN, likely due to its increased receptive field and model complexity. We believe this showcases nicely the merits of our work: Our operators can be easily used in more complicated propagation schemes and achieve even better results. This motivates further work that builds on our analysis. We are very thankful for that pointer!
>
>
>
> ## Weakness 2: Baselines
>
> As requested by other reviewers, we extend our evaluation by two additional baselines:
> - Following [1], we use the Laplacians of our work to create positional edge embeddings and feed them into a transformer (MagNet baseline).
> - We also follow the suggestion of another reviewer and consider a different mechanism to encode directed edges, following recent work on directed node-level problems [2] (DirGNN baseline).
> We report their performance in Tables 4,5,9-14 and find that EIGN outperforms both baselines.
>
> ## Weakness 3: Formatting
> - We add full stops to equations where needed.
> - We clarify that B^H means the conjugate transposed.
> - We capitalize Laplacian in the References.
>
> ## Question 1: Sign-Invariant Activation
>
> The tanh activation meets this criterion and is used in previous orientation equivariant architectures [4]. In general, any odd (and non-linear) function is possible (sin, sinh, sgn, ...).
>
> ## Question 3: Node Signals
>
> Yes, the boundary operators of our proposals define a natural mapping to and from the node-level domain. Each boundary map translates node signals to orientation equivariant/invariant signals respectively.
>
> ## Question 4: Weighted Signed Graphs
>
> We see no problem in applying this to weighted graphs as well: Instead of using values of magnitude 1 in the boundary operators, using $\pm w^{1/2}$ recovers a weighted Laplacian operator. Similarly, one can define independent Laplacian convolutions for positive and negative edges, thus also covering signed graphs. Similar to the DirGNN baseline [2], one challenge would be to ensure information exchange between both edge types. By using the Magnetic Laplacians for both mechanisms one could represent signed and directed edges simultaneously. The datasets we study, however, do not support this data modality. If the reviewer has a pointer for a scenario where this modeling paradigm is needed, we would be very curious to hear that. We add a pointer regarding this direction to the future work section of our manuscript.
>
>
> We again want to thank the reviewer for their helpful input. Especially the high efficacy of more complex propagation schemes using our Laplacians like a ChebNet showcases the merits of our study. If we left some points inadequately addressed we would be very happy about further discussions.
>
>
> ## References
> [1] Geisler, Simon, et al. "Transformers meet directed graphs." International Conference on Machine Learning. PMLR, 2023.
>
> [2] Rossi, Emanuele, et al. "Edge directionality improves learning on heterophilic graphs." Learning on Graphs Conference. PMLR, 2024.
>
> [3] Defferrard, Michaël, Xavier Bresson, and Pierre Vandergheynst. "Convolutional neural networks on graphs with fast localized spectral filtering." Advances in neural information processing systems 29 (2016).
>
> [4]: Roddenberry, T. Mitchell, Nicholas Glaze, and Santiago Segarra. "Principled simplicial neural networks for trajectory prediction." International Conference on Machine Learning. PMLR, 2021.

---

> > ### Comment · Reviewer_fGAT · 2024-11-25
> > **Raised Score**
> >
> > Thank you for your response. In light of your improvements, I have increased my score

---

### Official Review · Reviewer_mjjQ · 2024-11-03

**Soundness:** 3
**Presentation:** 3
**Contribution:** 3
**Rating:** 8
**Confidence:** 3

**Summary:**

The author proposes a novel model that decouples direction and orientation. The framework explicitly models equivariant/invariant orientation constraints for undirected and directed edges in graphs via different types of Laplacians. The model is comprehensively evaluated across a wide range of tasks and datasets, and outperforms prior works by a large margin.

**Strengths:**

The paper defines a clear problem to solve, and the solution is well-motivated. Specifically, decoupling direction from orientation and differentiating directed equivariant edge signals in different scenarios make the work unique compared to prior works. I believe the setup for controlling edges’ signals can be set as a new standard for the field. The theoretical analysis is also comprehensive and sheds light on the design choice. The experiments are also well-designed. They cover different tasks related to directed graphs with no parallel edges. The proposed novel set of benchmarks, which includes 3 synthetic datasets and 5 real-world datasets with 3 different tasks, is a significant contribution to the field.

**Weaknesses:**

One weakness of the paper is on the presentation side. The paper, even displays a lot of figures, to demonstrate different concepts, is still vague in many arguments. Please refer to the Questions section for detailed concerns. Another problem is model comparison. I think the paper will be more comprehensive if the model is compared with the more recent relevant work such as [1].  The next problem is the scalability problem. It seems to me that the novel way of different Laplacian constructions is a fairly computationally expensive process due to the unconventional way of defining incidence matrices. The benchmark itself also only contains small graphs, so it is unclear if the model can scale. Also, the method itself seems to not work on directed multigraphs. Intuition for using Magnetic Laplacian isn’t clear to me; do we really need to introduce complex phase shift to represent directed edges, or can we just pick an arbitrary value and separate undirected and directed Laplacian? Also please refer to question 5 for my concern.

[1] Geisler, S. et al. Transformers Meet Directed Graphs. (2023).

**Questions:**

1. On line 200-203, the authors state that “We restrict our novel constraints to undirected edges by requiring equivariance/invariance among direction-consistent orientations only. This allows representing the direction of directed edges through their orientation which, consequentially, must not be arbitrary.” Could you please explain the meaning behind this statement in a simpler term? If I understand correctly, orientation is generated arbitrarily for undirected edges, and direction-consistency only matters for directed edges. Also why does the representation of undirected edges allow representing the direction of directed edges?
2. Figure 2 should be explained more instead of just stating that the representations are indistinguishable for models that are orientation-equivariant for directed edges.
3. What is the time complexity to form the incidence and laplacian matrices in the paper?
4. I don’t think $\mathbf{B}_{\text{inv}}$ should be claimed to be a novel Laplacian as it is already standard definition in many works.
5. How sensitive the model is with respect to the hyperparameter $q$? It seems to me that if an edge has $k$ incoming edge messages such that $kq=2\pi m$ where $m$ is a natural number, the information regarding direction will be wiped out (assuming edge features are the same across edges).

---

> ### Author Response · Authors · 2024-11-21
>
> We thank the reviewer for their extensive review and are happy they find our paper clear, well-motivated, and unique from other works. We also agree that our evaluation including novel tasks is a significant contribution to the field.
>
> ## Weakness 1, Question 2: Presentation on Figure 2
> We discuss in the updated manuscript that the orientation can not encode edge direction as it is chosen arbitrarily. Models that are orientation equivariant for directed edges can not capture problem constraints that may arise in practice. We elaborate on this extensively in the new Appendix B.
>
> ## Question 1: Non-arbitrary orientation of directed edges
> This is an artifact of an earlier version of the paper and we removed it from our updated manuscript. The orientation assigned to directed edges plays no role for the boundary operators. From an implementation perspective, we overload orientation to encode edge direction. From a theoretical perspective, this is inconsequential and we apologize for the confusion.
>
> ## Weakness 2: Model Comparison
>
> We add two additional baselines:
> - Following [1], we use the Laplacians of our work to create positional edge embeddings and feed them into a transformer (MagNet).
> - We follow the suggestion of reviewer s2oj and consider a different mechanism to encode directed edges, following a node-level GNN [2] (DirGNN).
> Both are detailed in Appendix D.3 and thoroughly evaluated in Tables 4,5,9-14. EIGN outperforms both. Concerning MagNet [1], this approach does not adequately model equivariant signals and suffers from the same shortcomings as other fully orientation invariant baselines (MLP, LineGraph, ...).
>
> ## Weakness 3, Question 3: Scalability
>
> All Laplacians can be constructed without materializing boundary operators, as detailed in Table 7. The Laplacians are highly sparse and have as many non-zero entries as there are edge pairs that are connected through a shared node. Thus, EIGN has the same runtime complexity as all line-graph-based models. If the line graph has $F$ edges, the complexity of each layer is in $\mathcal{O}(F)$. We choose boundary operators in our paper as they theoretically motivate our approach and also provide some intuition.
>
> ## Weakness 4: Intuition for Magnetic Laplacian
>
> There are different approaches to distinguish directed and undirected edges. Using instead a real number to shift directed signals is less expressive: One could not distinguish the effect of a directed edge from a rescaled real input on an undirected edge. For the Magnetic Laplacian, since complex phase shifts are only induced by directed edges, the complex phase uniquely identifies directed edges. We use this concept as it has recently been shown to be very effective in encoding the topological structure of partially directed graphs on the node level [1].
>
>
> ## Question 4: Invariant Boundary
>
> Previous work on topological methods uses the equivariant boundary operator which is also sometimes referred to as a signed incidence matrix. The node Laplacian is constructed from this operator as well. To the best of our knowledge, the fully invariant boundary is not used in previous work. If the reviewer can point us to a paper that already uses this operator, we will adapt our manuscript accordingly.
>
> ### Question 5: Phase Shift q
>
> We provide an ablation regarding $q$ in Figure 7. Indeed, this hyperparameter needs to be chosen carefully as discussed in Appendix E. As correctly pointed out by the reviewer, too large values induce ambiguities or make the phase shift irrelevant altogether. We chose $q = 1/m$ to mitigate any issue from arising and justify this choice also empirically. Overall, both too small or too large values may result in worse model performance, while $1/m$ achieves satisfactory results over all datasets.
>
> We want to thank the reviewer for the very detailed questions. If our updated manuscript and additional experiments still leave some concerns unanswered we are happy for any additional pointers.
>
> ## References
> [1] Geisler, Simon, et al. "Transformers meet directed graphs." International Conference on Machine Learning. PMLR, 2023.
>
> [2] Rossi, Emanuele, et al. "Edge directionality improves learning on heterophilic graphs." Learning on Graphs Conference. PMLR, 2024.

---

> ### Comment · Reviewer_mjjQ · 2024-11-25
>
> **Question 4**:
> I think these works [1,2,3] use the notion of upper/lower adjacency, which is very similar to your $ L_{inv}=B_{inv}^T B_{inv} $ with the diagonal set to 0. Another work [4] also constructs a similar notion under the name of down Laplacian matrix. You can take a look at their official repository, where they have few examples as well.
>
> Given that all of the concerns are addressed, I will raise my score. Thank you very much for your thoughtful response.
>
> [1] Bodnar et al., Weisfeiler and Lehman Go Topological: Message Passing Simplicial Networks, ICML'21.
>
> [2] Bodnar et al., Weisfeiler and Lehman Go Cellular: CW Networks, NeurIPS'22.
>
> [3] Truong and Chin, Weisfeiler and Lehman Go Paths: Learning Topological Features via Path Complexes, AAAI'24.
>
> [4] Hajij et al., TopoX: A Suite of Python Packages for Machine Learning on Topological Domains.

---

> > ### Author Response · Authors · 2024-11-27
> >
> > Thank you for these pointers. While these works (implicitly) use and implement the invariant (unsigned) boundary operator, they neither explicitly construct Laplacians nor do they consider invariance as a desirable property. We adapted our manuscript to reference the relevant works accordingly (L. 292).
> >
> > We are happy to have been able to address your concerns. Thank you for the useful input!

---

### Official Review · Reviewer_s2oj · 2024-11-03

**Soundness:** 2
**Presentation:** 3
**Contribution:** 3
**Rating:** 6
**Confidence:** 4

**Summary:**

This paper focuses on dealing with edge values on graphs that can only be defined up to an orientation assigned to that edge. Such values flip their sign if the orientation direction of that edge is changed. Conversely, there are edge values that are scalars and do not require an orientation to be defined. Functions that aggregate and update edge values can be either equivariant (the predicted value changes with orientation) or invariant (the predicted value remains the same) to changing the orientation of edges. The authors propose GNN model that at each layer takes in as input orientation-aware equivariant and invariant edge values and outputs updated orientation-aware equivariant and invariant edge values. The authors use a clever design to ensure that the model keeps the equivariant and invariant edge values consistent across the layers. They also introduce a principled way to aggregate across equivariant and invariant edge values. Finally, they extend they approach to graphs with directed edges by using a magnetic Laplacian. Directed edges always have a well defined orientation given by the edge direction. The magnetic Laplacian incorporates a complex phase shift to values of directed edges during the aggregation step. The authors use synthetic data and some real world data sets that their proposed method outperform existing methods on graphs with orientation-aware edge values.

**Strengths:**

This reviewer generally liked the paper. The reviewer was not aware of the problem of graphs whose edge can take on both orientation-aware equivariant and invariant values. However, the author's real world data sets seem convincing and not too contrived. It seems like there is a problem here to solve. The way the authors construct their framework, for example how the equivariant and invariant graph Laplacians are constructed using boundary operators and stitched together, is clever and principled. There is no argument that the proposed GNN respects the symmetries of the problem when it comes to equivariance and invariance of edge values with respect to edge orientations. The empirical evidence is also strong. It seems that respecting the symmetries does help with performance. Finally, the authors do a reasonable job with their ablation studies to highlight where the improvements form their method is coming from.

**Weaknesses:**

A minor weakness is potentially the nuanced nature of this problem. To this reviewer the problem of applying GNNs to graphs with mixed undirected and directed edges and edge values on undirected edges that can be both equivariant and invariant to their orientation could be viewed as too much of a niche problem and perhaps not enough of a significant advance for the field of learning of graphs more broadly. Therefore, this paper may lack broad interest at ICLR.

The main weakness of the proposed method is the use of magnetic Laplacian and phase shifts in the complex plane to account for directed edges. Although magnetic Laplacian is a reasonable way to aggregate information across neighboring nodes in a directed graph, it is not the most general way to do so and has limited expressivity. The reason for this is simple. The phase used to weigh features across directed edges can result in information loss when the complex features are aggregated at each node. For example, a phase angle of q aggregated with a phase angle of -q results in real value that is indistinguishable from a value aggregated from undirected edges. Similarly if q = 1/2 then 4 hops along directed edges will result in a purely real value indistinguishable from messages aggregated from undirected edges. Similarly, after multiple hops where the edge values may become complex, it will not be possible to determine whether a complex value arriving at a node came from a real value propagated along a directed edge or a complex edge value sent along an undirected edge. Therefore, the use of a phase shift and complex numbers is not the most general way to aggregated information in directed graphs and results in loss of information and thereby lower expressivity of GNNs.

Arguably, the most general way to do aggregation of features of neighboring nodes in a directed graph is proposed by Rossi et al. There, the features from neighbors connected by outgoing edges is aggregated separately from that of neighbors connected by incoming edges. This can be easily extended to include undirected edges as a separate category. During the update step the aggregated features of outgoing and incoming neighbors are concatenated alongside the self feature of the node. The authors should consider such a scheme and construct their boundary operator this way (a vector now that separately aggregates features from outgoing and incoming directed edges and undirected edges). The Laplacian operator can be constructed using this boundary operator and the same constraints applied to ensure that equivariance and invariance requirements are kept. Given how principled the author's approach has been to respecting the symmetries with respect to edge orientation, it would be interesting to see if the most principled way to incorporate edge directions will further improve the performance of their model.

**Questions:**

Please see comment above regarding weaknesses. Can the authors comment on whether it is possible to do away with magnetic Laplacian and construct their method using the most general framework for aggregating information in directed graphs from Rossi et al.? If so, can the authors implement this approach and do an empirical comparison to their current approach
using the magnetic Laplacian?

To address this reviewer's concern about the niche nature of the problem tackled in the paper, could the authors elaborate on the broader applications of their work? What sort of real world problems can be solved using the proposed approach? Are there more general implications of the proposed approach for solving related problems?

---

> ### Author Response · Authors · 2024-11-21
>
> We thank the reviewer for their thorough and elaborate review and are happy that they, too, find the problem that our work studies to be important and well-addressed by our paper.
>
> ## Relevance to ICLR and Impact
>
> We strongly believe that this work is relevant to the community. Recently, there has been a growing interest in topological methods for various settings [1-5]. All of these assume equivariant signals for edges exclusively whilere there are basically no real-world applications which meet these strict assumptions. Consequently, this comes at the cost of greatly sacrificing model performance. While there are no established benchmarks for settings with equivariant and invariant signals, we believe that our paper bridges a relevant gap toward practical applications:
>
> Real-world settings almost always come with both signal modalities. Domains include traffic and electrical engineering - as studied in our paper - but also hydraulics, water flow networks, electrical grids, pneumatics, engineering (statics, as buildings could be modelled as graphs) where there are yet no benchmarks available. In practically all aforementioned applications, orientation-invariant signals are crucial: Pipe diameter, material properties, etc. heavily influence the nature of the problem. Critically, none of the existing works (we highlight [1-4], but many more exist) can be effectively applied to any of these settings because of their restrictive assumptions - as is supported by our evaluation.
>
>
> ## Magnetic Laplacians
>
> We appreciate the concerns regarding the Magnetic Laplacians and the example-based justification by the reviewer. We think that investigating a propagation scheme according to [6] is a good suggestion and we thank them for the detailed description. Nevertheless, while we do not claim that the Magnetic Laplacian is as general as [6], neither the theoretical arguments are complete nor do we find that the proposed approach outperforms EIGN. We elaborate on the theoretical arguments in detail below but, in short, the argument neglects the existence of the additional learnable mappings. Due to the learnable mappings, it is hard to argue about phase shifts canceling out as the model can learn to mitigate this behavior. Without learnable mappings, [6] also can not distinguish the aggregation for directed and undirected edges through different mappings.
>
> We included a comparison with [6] and detail this additional experiment (Appendix D.3, L. 1353). We construct three different boundary operators for each modality: One for undirected edges, one for incoming edges, and one for outgoing edges. Each of them is associated with a separate trainable weight matrix in the adopted version of EIGN. We add the resulting baseline (DirGNN) to all our evaluations (Tables 4,5,9-14) and conduct the same hyperparameter search as for all models.
>
> Even though this baseline satisfies joint orientation equivariance and invariance, EIGN still outperforms. One reason may be the sparsity structure of the corresponding Laplacians: The first will only propagate information between undirected edges only, while the other two will only propagate information between edges that are incoming/outgoing regarding a shared node. This heavily limits the expressivity of these operators as information between directed and undirected edges can only be exchanged over multiple layers. Note that all three message-passing schemes combined recover only a subset of the adjacency structure of our Laplacians. In Table 7, only the third and the last two lines are represented by this baseline. Furthermore, this high sparsity limits the capability of models to predict orientation equivariant targets from only orientation invariant features which is done through inter-modality Laplacians (see L. 489). In simulation tasks, the baseline falls back to predicting mostly zeros as well.
>
> Again, we do not want to make any claims as to which approach is "the most general" to represent directed edges. Many node-level models can not easily be transferred to the edge level without severe limitations. The Magnetic Laplacian is one framework that recently saw success and can be applied rather elegantly. In our updated manuscript (L. 528), we acknowledge different strategies that can also benefit from the theoretical groundwork regarding equivariance and invariance.
>
> We also want to highlight that how directed edges are differentiated is only one part of our contribution: Formalizing and satisfying the joint orientation equivariance and invariance desiderata is the key to effectively addressing general edge-level problems.

---

> ### Author Response · Authors · 2024-11-21
>
> Finally, to address the reviewer's examples:
>
> 1. Indeed, for the Magnetic Node Laplacians, the complex parts of signals can cancel out but remain largely distinguishable. Consider two very simple graphs of undirected and directed edges respectively: `v1 - v2 - v3` and `v1 -> v2 -> v3`. In the case of only uniformative node features (constant 1), for the undirected case `v2` will have $1 - \sqrt{2} - \sqrt{2}$. For the directed case, the `v2` will have $1 - \sqrt{2}exp(i\pi q) - \sqrt{2}exp(-i\pi q) = 1 - 2\sqrt{2}cos(\pi q)$. Both are real, but not the same and, thus, distinguishable.
> 2. For a sequence of directed nodes `v1 -> v2 -> ... -> vn`, the information of `v1` will be shifted by $nq\pi$ when reaching `vn` without reaching an MLP which may nullify the phase shift. However, we chose $q=1/m$ to bound the maximum path length which prevents this from happening. Again, since in practice learnable transformations are used, a model can learn to mitigate this cancellation. We also ablate the choice of $q$ in Figure 7.
> 3. For a sequence `v1 -> v2 - v3 - ... - vn`, the information of `v1` that reaches `vn` does not encode at which point in the sequence a directed edge occured only if no learnable mappings are used. In practice, the model can simply learn to keep track of how many hops information "has travelled" and therefore distinguish the example graph from `v1 - v2 -> v3 - ... - vn`.
>
> We thank the reviewer for their insightful comments, in particular for their alternative proposal to encode edge direction. We believe this comparison supports Magnetic Laplacians to encode directionality. We also hope that we were able to motivate why our paper is placed well in the current literature. If any concerns remain we would be very happy to discuss them further.
>
> ## References
>
> [1]: Roddenberry, T. Mitchell, and Santiago Segarra. "HodgeNet: Graph neural networks for edge data." 2019 53rd Asilomar Conference on Signals, Systems, and Computers. IEEE, 2019.
>
> [2]: Roddenberry, T. Mitchell, Nicholas Glaze, and Santiago Segarra. "Principled simplicial neural networks for trajectory prediction." International Conference on Machine Learning. PMLR, 2021.
>
> [3]: Bodnar, Cristian, et al. "Weisfeiler and lehman go topological: Message passing simplicial networks." International Conference on Machine Learning. PMLR, 2021.
>
> [4]: Schaub, Michael T., et al. "Signal processing on higher-order networks: Livin’on the edge... and beyond." Signal Processing 187 (2021): 108149.
>
> [5]: Papamarkou, Theodore, et al. "Position: Topological Deep Learning is the New Frontier for Relational Learning." arXiv preprint arXiv:2402.08871 (2024).
>
> [6]: Rossi, Emanuele, et al. "Edge directionality improves learning on heterophilic graphs." Learning on Graphs Conference. PMLR, 2024.

---

> > ### Comment · Reviewer_s2oj · 2024-11-28
> >
> > I thank the authors for their comprehensive response. I respectfully disagree that learnable transformations at each node alleviate the ambiguity introduced by the magnetic Laplacian. Even in the simple example 1 provided by the authors, it is clear that for certain values of q (q=2 in this case for example) the resulting propagated features will be the same for the undirected and directed graphs. It is true that a prudent choice of q will resolve this ambiguity as pointed out in point 2 by the authors (and the added discussion in the revised text). The issue is that both the q values and the learnable transformations are the same across all the nodes of the graph, whereas resolving the ambiguity caused by the magnetic Laplacian will depend on the details of the propagation and will be in general different for different nodes. Therefore, a prudent choice of q and learning a transformation at each node will not generally alleviate the problem. Nevertheless, I commend the authors for trying a DirGNN type architecture for their approach and reporting the empirical results. I agree with the authors that the main point of their paper is not the directed graph implementation but rather the joint equivariance and invariance formulation. I will therefore increase my score.

---

### Official Review · Reviewer_X95w · 2024-11-04

**Soundness:** 4
**Presentation:** 4
**Contribution:** 2
**Rating:** 6
**Confidence:** 4

**Summary:**

This paper distinguishes between orientation invariant (i.e. those with fixed sign regardless of orientation) and equivariant (those that change sign depending on orientation) features. It highlights the importance of being able to model both types of features in a graph learning scenario and proposes a new model that is able to do so.

**Strengths:**

The paper is well-written, bringing across the main point nicely. The theoretical claims are sound and the proofs seem correct.

**Weaknesses:**

The main idea of the paper brings little novelty. Both invariant and equivariant edge features have been explored previously - just not together. The performance gain could well be attributed to a more complex model.
The theoretical contributions are very simple. Essentially Theorem 4.1 follows by design.

**Questions:**

- How many trainable parameters do the models use respectively?
- How sensitive to hyperparameters is your proposed model?

---

> ### Author Response · Authors · 2024-11-21
>
> We thank the reviewer for their review and are happy they like the presentation and our perspective on the importance of different signal modalities.
>
> ## Novelty
> The key contribution of our paper is jointly modeling both edge-level modalities. Since prior work [1-3] does not handle with and without orientation simultaneously, it is highly ineffective in practically all applications. While each modality has been discussed in prior work individually, to the best of our knowledge, no work even mentions this issue/limitation that arises in essentially all applications that contain orientation equivariant signals. Already discussing this issue is a valuable contribution to the field. This is also acknowledged by the other reviewers: "clever design" (s2oj), "work unique compared to prior works" (mjjQ), "non-trivial" (fGAT).
>
> Combining the modalities is non-trivial: Previous notions of equivariance must be adapted and edge direction must be modeled in a principled way, which is also a novel contribution. Our empirical results strongly underline this point: Simply relying on existing work and combining both signal types performs significantly worse.
>
> We conduct additional experiments to show that the success of EIGN is not due to its complexity but its inductive biases that come from our theoretical considerations:
> - Two new baselines: A transformer with direction-aware positional encodings, and a GNN similar to EIGN that uses a different mechanism to encode edge direction. The transformer has more parameters than EIGN (see Table 17).
> - Even though EIGN and the baselines were tuned over the same hyperparameter space, we ablate variants of the baselines with larger hidden dimensions to match EIGN's number of parameters.
>
> Table 17 of our updated manuscript shows that EIGN still significantly outperforms. If the reviewer still views this experiment as insufficient evidence for the merits of our theoretical analysis and resulting model, we would be very happy for pointers to additional experiments we can conduct.
>
> ## Theoretical Contribution
>
> We first formalize edge-level problems from a theoretical perspective and propose suitable desiderata - joint orientation equivariance and invariance in the presence of directed and undirected edges. While already the formalization is a novel aspect that due to the negligence of prior work is an important contribution on its own, designing operators that meet the requirements is highly non-trivial. Especially, the interactions $L_{inv \rightarrow equi}$ and $L_{equi \rightarrow inv}$ have no comparable equivalent in the literature. Naturally, we design these operators with the clear goal of satisfying the novel equivariance and invariance conditions while being flexible enough to be used in future work. For example, we show that our Laplacians can be used in sophisticated message-passing schemes like Chebyshev polynomials, as requested by reviewer fGAT, which exemplifies the utility of our Laplacians beyond our work.
>
> We want to stress that our theoretical contribution spans multiple aspects: We identify and formally address shortcomings of previous concepts (Definitions 4.1-4.2), and develop suitable operators to address them. Theorem 4.1. follows from carefully designing Laplcians that satisfy the new desiderata. Essentially, we split the otherwise lengthy proof into three parts, Lemmata 4.1-4.3., and Theorem 4.1. follows from their composition.
>
> ## Q1: Number of Trainable Parameters
>
> We report the number of parameters for all baselines and larger variants (see above) in Table 17. EIGN outperforms all baselines even when they have a similar number of parameters.
>
> ## Q2: Sensitivity to Hyperparameters
>
> We thank the reviewer for pointing this out. We provide two additional ablations regarding the hyperparameters of EIGN:
> - Figure 7 shows the sensitivity to the phase shift q and discusses appropriate choices from a theoretical perspective.
> - Table 16 shows EIGN's performance among the hyperparameter configuration space it was tuned: For a large range of values, EIGN outperforms all baselines.
>
>
> Overall, we are very grateful for these questions. If we did not fully resolve the reviewer's concerns, we would greatly appreciate further clarification.
>
> ## References
>
> [1]: Roddenberry, T. Mitchell, and Santiago Segarra. "HodgeNet: Graph neural networks for edge data." 2019 53rd Asilomar Conference on Signals, Systems, and Computers. IEEE, 2019.
>
> [2]: Bandyopadhyay, S., K. Das, and M. N. Murty. "Line hypergraph convolution network: Applying graph convolution for hypergraphs. arXiv 2020." arXiv preprint arXiv:2002.03392 (2002).
>
> [3]: Geisler, Simon, et al. "Transformers meet directed graphs." International Conference on Machine Learning. PMLR, 2023.

---

> > ### Comment · Reviewer_X95w · 2024-11-26
> >
> > Dear Authors,
> >
> > Thank you for providing answers to my questions. Taking into account the insight you have provided, I have updated my score.

---

### Author Response · Authors · 2024-11-21

We thank all reviewers for their thorough and helpful feedback. We **updated our manuscript** to address questions. We highlight additions and **changes in blue**. Additionally, we provide more experiments and ablations:

- We ablate the hyperparameters of our model, in particular $q$. This nicely supplements the theoretical motivation for setting $q=1/m$.
- We adopt two node-level baselines for edge-level problems: A transformer with direction-aware positional encodings and a message-passing scheme that separates directed and undirected edges. EIGN outperforms both, underlining the merits of our modeling paradigm.
- We show that our principled Laplace operators are also great building blocks for more sophisticated convolution operators. Specifically, we follow reviewer fGAT's request and parametrize Chebyshev polynomials with our Laplace operators. This showcases the utility of the principled Laplace operators and the proposed notions of equivariance and invariance.

Overall, we believe all of these experiments help to make our main point even more convincing: Different edge-level signal modalities must be treated carefully, and our work lays the theoretical groundwork for a holistic approach to edge-level problems. Our empirical evaluation contributes an appropriate benchmark and highly flexible design principles that consistently outperform previous work. None of the existing prior works can properly handle different signal modalities, which renders them effectively useless for virtually every practical application. We address and resolve this shortcoming to facilitate a broad applicability of graph machine learning for edge-level signals.

---

### Meta-Review · Area_Chair_EJSn · 2024-12-11

**Metareview:**

This paper extends equivariant ML to directed signals on directed graphs. All reviewers agreed this paper is worthy of publication in this venue. Some reviewers emphasized the simplicity of the results and that the problem is well-motivated by real-world scenarios. Others mentioned that the solution was elegant, while others valued the thorough ablation in the numerical experiments.

**Additional Comments On Reviewer Discussion:**

One reviewer raised their concern that the paper had little novelty and the theoretical claims were obvious. However, after the discussion period, the reviewer raised their score to 6.

---

### Decision · Program_Chairs · 2025-01-22

Accept (Poster)